# Learning of distant state predictions by the orbitofrontal cortex in humans

G. Elliott Wimmer [1,2] & Christian Büchel [3]

Representations of our future environment are essential for planning and decision making. Previous research in humans has demonstrated that the hippocampus is a critical region for forming and retrieving associations, while the medial orbitofrontal cortex (OFC) is an important region for representing information about recent states. However, it is not clear how the brain acquires predictive representations during goal-directed learning. Here, we show using fMRI that while participants learned to find rewards in multiple different Y-maze environments, hippocampal activity was highest during initial exposure and then decayed across the remaining repetitions of each maze, consistent with a role in rapid encoding. Importantly, multivariate patterns in the OFC-VPFC came to represent predictive information about upcoming states approximately 30 s in the future. Our findings provide a mechanism by which the brain can build models of the world that span long-timescales to make predictions.

[1] Max Planck University College London Centre for Computational Psychiatry and Ageing Research, London WC1B 5EH, UK. [2] Wellcome Centre for Human Neuroimaging, University College London, London WC1N 3AR, UK. [3] Department of Systems Neuroscience, University Medical Center Hamburg-Eppendorf, Hamburg 20246, Germany. Correspondence and requests for materials should be addressed to G.E.W. (email: elliott@caa.columbia.edu)

Making predictions about the future has been proposed to be a core organizing principle of brain function[1–3]. Over short timescales, sensory and motor systems enable the prediction of immediate stimuli and of the consequences of our actions[1,4,5]. Over longer timescales, predictions about the future state of the world—and the associated positive and negative consequences of our potential actions—are essential for goal-directed decision-making and planning for the future. While the neural mechanisms of prediction in sensory and motor systems have been the subject of extensive research, we know little about the neural systems involved in learning to make predictions from experiences that extend beyond several seconds.

Neural systems including the hippocampus and the orbitofrontal cortex (OFC) have emerged as potential substrates for representing both spatial and abstract maps of the environment[6–8]. Building on the original discovery of place representations in hippocampal neurons[9], the hippocampus has been proposed to encode relations between stimuli that can combine to form a cognitive map of the environment[7]. Supporting a role for hippocampal representations in active prediction, research in animals has shown that when freely navigating an environment, activation of "place cell" sequences can predict future movement trajectories[10,11]. In humans, the hippocampus has also been shown to support the reactivation of visual associations originally separated by a few seconds[12]. Other fMRI studies have found evidence for visual cortex reactivation for similar short-timescale associations[13,14], however, especially over longer timescales, such predictive knowledge must be supported by a still undetermined learning system outside of the sensory cortex[5].

Research on the orbitofrontal cortex has recently found support for related cognitive functions as the hippocampus. While the OFC has long been viewed as critical for value representation and flexible learning, a unifying account proposes that the OFC represents a cognitive map of relevant states in the environment[6,15–18]. For example, using fMRI in humans, Schuck et al.[15] demonstrated that the OFC represented recently observed hidden contextual information that was necessary for task performance. Interestingly, the results from a recent study in rodents indicate that the OFC and the hippocampus interact to represent task-relevant structure and guide behavior[19,20]. However, it is not known whether and how the hippocampus and/or OFC acquire and represent predictions during goal-directed behavior.

From a computational perspective, learning longer timescale predictions poses difficulties for reinforcement learning models that track only the value of states. Temporal difference learning models are constrained by the slow propagation of errors back through preceding states[21], and neurally, the reward prediction error signal carried by midbrain dopamine neurons loses its fidelity beyond 5–10 s after feedback[22,23]. Alternative computational models that increase learning speed have been proposed, including eligibility traces[21] as well as models that explicitly represent state transitions, such as successor representations and model-based learning[24–26]. Positive evidence for learning of long-timescale prediction in the hippocampus or OFC would support a neural mechanism underlying model-based or successor representation accounts of learning in multi-stage decision problems[27–29].

To investigate learning mechanisms supporting the prediction of distal future states, we focused on initial learning as participants navigated eight unique Y-maze environments, each of >40 s duration. Participants learned to select the correct left or right door in an initial room (state 1) that led down one of two arms to a delayed reward (versus a loss) after moving through two intervening forced-choice states (Fig. 1a, c). Critically, state 3 was represented by a unique and different category stimulus than state 1, allowing us to examine the development of predictive representations of state 3 when participants make their choice at state

1. The four repetitions of each maze were separated by at least 4 min, a delay that minimizes short-term maintenance of choice strategy and likely requires a contribution of longer-term memory for successful learning[30,31]. By studying the initial learning process in relatively long-duration experiences, our goal was to better understand learning in situations like those outside the lab where rapid learning is evident after only one or a few experiences.

Using fMRI in humans, we find that activity in the hippocampus is higher during initial exposures to each maze, potentially reflecting rapid encoding of a representation of the environment. Critically, across learning we also find that predictions of distal states ~30 s in the future come to be reflected in patterns of activity in the medial OFC/ventromedial PFC (OFC-VMPFC).

## Results

**Behavioral results**. In the navigation learning experiment, behaviorally, participants showed a rapid increase in mean performance across repetitions of the eight unique mazes, rising to 76% correct at the second exposure (CI [70.0, 83.6]; $t_{(34)} = 22.81$, $p < 0.0001$, $t$ test versus chance; Fig. 1b). Initial feedback also exerted an effect on learning: if a reward was received on the first exposure, performance on the next repetition was significantly higher than performance following an initial loss (repetition 2, initial reward, 82.1% CI [76.0, 88.2]; $p < 0.0001$, $t$ test; repetition 2, initial loss, 71.4% (CI [62.1, 80.8]; $p < 0.0001$, $t$ test; $t_{(34)} = 2.77$, CI [2.8, 18.6]; $p = 0.009$, $t$ test). Supporting the involvement of multiple cognitive systems in learning, we found that individual differences in working memory capacity as measured by the OSPAN task significantly correlated with mean accuracy over the learning phase ($r = 0.39$, $p = 0.026$, Pearson correlation; OSPAN performance mean 38.3, range, 0–86; arithmetic component performance mean 89.2%). The relationship between performance and working memory is likely related to the maintenance of state 1 information throughout the long delay until feedback; in contrast, it is unlikely to be related to working memory maintenance of choice policy across repetitions, as repetitions of the eight individual mazes were separated by 4 min on average. While episodic memory and working memory are often viewed as a distinct functions, the hippocampus is also known to be involved in critical aspects of working memory maintenance[32].

**fMRI univariate results**. Our fMRI analysis first examined univariate learning-related brain responses. At the first state in a maze, state 1, we expected to find a learning signal reflected in an initially high and then (exponentially) decreasing response across repetitions in regions supporting memory and associative encoding. We indeed found a significant decreasing effect of exposure number (repetition) on BOLD activity in the bilateral hippocampus (right: $z = 4.42$; $p = 0.004$ SVC; 18 −10 −21; left: $z = 4.05$; $p = 0.02$ SVC; −18 −14 −21; Fig. 2a, Supplementary Fig. 1), as well as the right parahippocampal gyrus ($p < 0.05$ whole-brain FWE-corrected; Supplementary Table 1). This effect was relatively selective, with only three other above-threshold clusters of activity in this contrast: two in the occipital lobe and one in the left dorsal premotor cortex (whole-brain FWE-corrected; Supplementary Table 1). The pattern of decreasing activity across repetitions was also selective to state 1; we found only a single significant cluster in the precuneus at state 2 and no significant clusters of activity in state 3 (Supplementary Table 1). Overall, the pattern of initially high and then decreasing activity over time in the hippocampus and parahippocampal cortex supports a role for the hippocampus and broader MTL in rapid episodic or relational learning.

Next, we confirmed expected responses to delayed reward feedback. We found that reward versus loss feedback, arriving

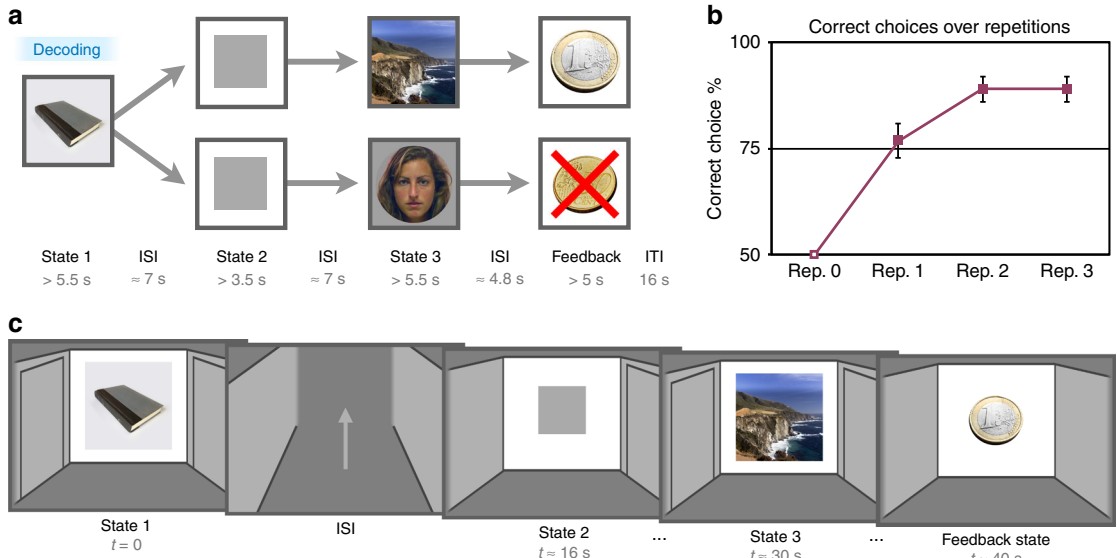

**Fig. 1** Learning-phase task design and behavioral results. **a** Learning-phase maze trial structure. Across eight unique maze sequences, participants learned which action to make in state 1 in order to receive a deterministic reward at the end of the maze. Participants made a left or right choice in the first state and then proceeded through instructed left or right choices in state 2 and state 3, followed by reward or loss in the feedback state. Maze-unique stimuli from three categories (faces, scenes, and objects) were presented in state 1 and state 3. The delay between state 1 and state 3 was on average greater than 30 s, while repetitions of unique mazes were separated by at least 4 min. Critical decoding analyses focused on representations of information about state 3 at the onset of state 1 (represented by "Decoding" in the blue box). **b** Mean learning-phase performance across four repetitions of each maze. Half of initial repetitions ended in reward and half in loss, and the resulting mean 50% level of performance is indicated by the open square at repetition 1. Error bars represent standard error of the mean. **c** Illustration of the participant's screen view, a cartoon room with potential left and right door options followed by a path through a hallway between states. In state 2 and state 3, the instructed choice was indicated by a shift in the central stimulus to the instructed door. A localizer phase, which was used to derive classifiers for faces, scenes, and objects, followed the learning phase

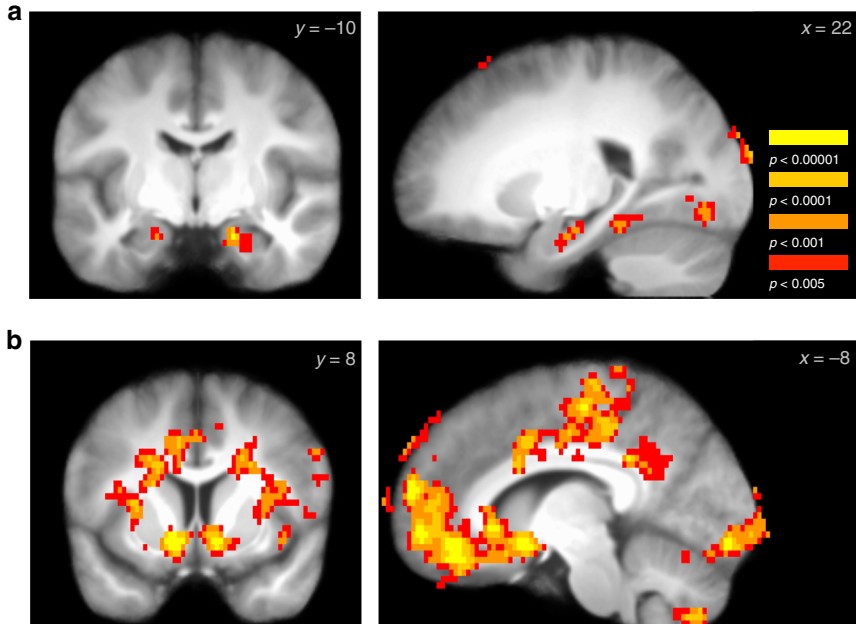

**Fig. 2** Learning-related univariate fMRI responses. **a** Across repetition of mazes during the learning phase, activity in the hippocampus significantly decreased, supporting a role for the hippocampus in maze encoding (right and left hippocampus, $p < 0.05$ SVC; image threshold $p < 0.005$ unc.). **b** At feedback, beginning >40 s after the start of a maze, activation in the ventral striatum and OFC - ventromedial PFC, among other regions, was significantly activated by the receipt of a reward versus loss. (Images $p < 0.05$ whole-brain FWE-corrected; full maps available at: https://neurovault.org/images/100616/ and https://neurovault.org/images/100619/)

~40 s after the state 1 choice for each maze, was correlated with BOLD activity in a cluster including the ventral striatum and OFC-VMPFC (Fig. 2b; Supplementary Fig. 2). We also found reward-related activation in the left hippocampus ($z = 3.84$; $p =$

0.028 SVC; $-30 -18 -15$; this effect was also part of a larger significant cluster in the whole-brain analysis), in line with predictions from fMRI research on responses to relatively delayed feedback (~7 s)[33]. Note that these results revealed some clusters of

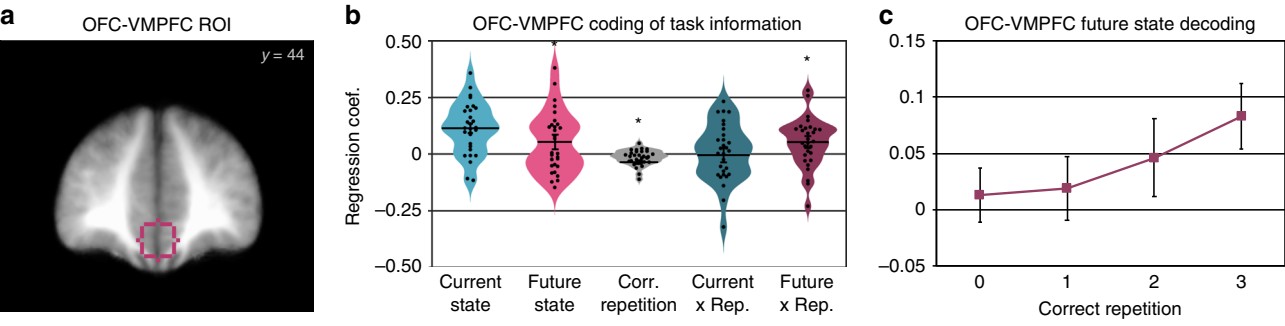

**Fig. 3** Multivariate OFC-VMPFC responses at state 1 onset related to the representation of anticipated future states. **a** Outline of the OFC-VMPFC region of interest shown over the across-participant average anatomical MRI. **b** Regression coefficients for decoded activity in the OFC-VMPFC for the current state (state 1), future state (state 3), the repetition since performance was correct, the interaction of current state with correct repetition, and the critical interaction of future state with correct repetition. Distribution density is represented by violin plot width; individual participant datapoints are in black. (*$p <$ 0.05, $t$ test; error bars represent standard error of the mean; statistical comparisons were not completed on the current state representation in (**b**) because the plotted data represent the results after exclusion of two participants with poor decoding during learning.) **c** Breakdown of the future state by correct repetition interaction, for illustration. Future state classification is plotted separately for each correct maze repetition; these effects were derived from control models examining current and future state separately for each correct repetition bin

apparent activation on the edge of the grey matter which could be due to residual noise or motion; it is possible that such effects could be removed with more advanced motion- and noise-correction algorithms[34,35]. Interestingly, we found that neural processing related to the effect of a reward versus loss persisted through the 16-s rest period following feedback. During rest, feedback continued to differentially affect activity in a cluster including the OFC-VMPFC and ventral striatum, as well as two clusters in the right posterior hippocampus (OFC-VMPFC and ventral striatum, $p < 0.05$ whole-brain FWE-corrected; right hippocampus $z = 4.75$; $p = 0.001$ SVC; 32 −34 −3; $z = 4.22$; $p = 0.007$ SVC; 30 −18 −12; Supplementary Fig. 2, Supplementary Table 1).

**fMRI multivariate results**. We next turned to multivariate analysis methods to pursue our primary question of whether learning mechanisms in the OFC-VMPFC and hippocampus acquire predictive information about future states. We examined an OFC region of interest based on Schuck et al.[15] (Fig. 3a) and a hippocampal ROI based on an anatomical atlas. As a control, we also analyzed responses in visual category-selective regions in the occipital and ventral temporal cortex.

We first verified that a classifier trained on multivoxel patterns of activity from the post-learning localizer phase was able to generalize to accurately classify stimuli at state 1 and state 3 during the learning phase (state 2 did not contain any image stimuli; Fig. 1a). Averaging across the three categories (faces, scenes, and objects), we found significant classification in all regions of interest (OFC-VMPFC mean AUC 9.88 CI [6.45,13.31]; $t_{(30)} = 5.88$, $p < 0.001$, one-sample two-tailed $t$ test versus zero; hippocampus 10.63, CI [7.68,13.57]; $t_{(30)} = 7.37$, $p < 0.001$, $t$ test; visual ROIs 48.21 CI [47.61, 48.85]; $t_{(30)} = 158.42$, $p < 0.001$, $t$ test). The classification of current-state information in the OFC-VMPFC was comparable or higher than previous reports (e.g., Schuck et al.[15]). For interpretation, it is important to note that conducting statistical inference on informational measures derived from cross-validation is problematic (see the Methods section)[36]. However, our classification decision values are derived from a different method, cross-classification, where training is conducted on the post-task localizer and testing on the learning phase. Moreover, because our learning effect of interest tests for changes in information over time, our primary measures are completely isolated from this concern.

We then used the trained classifier to test for evidence of prospective future state representation at the onset of state 1 in each maze, ~30 s before the expected future state became visible. Our method tests how well patterns of activity at state 1 were related to the actual current stimulus category on the screen, as well as the future state 3 stimulus category. The future state 3 information would be initially experienced upon the first rewarded repetition of a maze. In our analyses, we refer to this effect as "correct repetitions", which is coded from 0 (no experience with the future state) to 3 (maximum possible experiences with the future state). Using raw classifier decision values derived from patterns of BOLD activity at state 1, which reflect the strength of evidence for each tested category, we analyzed whether decision values were explained by the current state (state 1) and future state (state 3), and whether this changed with learning across repetitions (see Methods). Note that the use of a regression analysis provides a measure of effects versus null (zero), but does not yield a percent correct measure as reported in many classification studies. Importantly, the application of the category classifiers to the learning phase also will not be affected by any learned associations between reward and the stimuli. The three categories appeared (at state 1 and state 3) in rewarded and non-rewarded paths in approximately equal numbers, which would lead to similar value associations across categories after learning. Furthermore, any change in representation across learning cannot by definition be due to the applied classifier which is the same for all repetitions.

As expected, distributed patterns of activity in the OFC-VMPFC accurately discriminated the stimulus visible at the current state (including all participants; 0.092 CI [0.050, 0.134]; $t_{(30)} = 4.48$, $p < 0.001$, $t$ test; in this test and all following, one-sample two-sided $t$ tests were used; Fig. 3b). We found an effect of future state overall (0.052 CI [0.001, 0.103]; $t_{(28)} = 2.08$, $p = 0.047$, $t$ test; in this and following results, analyses exclude participants with poor current-state classification performance), although this effect is difficult to interpret. We also found a negative effect of correct repetition (−0.016 CI [−0.029, −0.003]; $t_{(28)} = −2.48$, $p < 0.02$, $t$ test). Note that the above caveat about interpreting main effects of across-phase decoding also applies to these effects of current and future state.

Critically, we found a significant increase in the representation of the future state across repetitions in the OFC-VMPFC (interaction between future state and correct repetition 0.051 CI [0.01, 0.093]; $t_{(28)} = 2.53$, $p = 0.017$, $t$ test; Fig. 3b). With a

medium effect size ($d = 0.47$ CI [0.01, 0.85]), a replication of the OFC-VMPFC result aiming for 80% power to detect an effect would require a minimum of 38 participants (two-tailed, uncorrected; $n = 30$, one-tailed). This effect of future state representation was selective, as the interaction between current state and correct repetition was near zero (0.007 CI [−0.041, 0.055]; $t_{(28)} = 0.30$, $p = 0.77$, $t$ test). The current state by repetition effect was also statistically equivalent to a null effect, as indicated by an equivalence test using the TOST procedure[37]: the effect was within the bounds of a medium effect of interest (Cohen's $d = \pm 0.55$, providing 80% power with 29 participants in the current analysis; $t_{(28)} = -2.66$, $p = 0.007$), allowing us to reject the presence of a medium-size effect.

In the hippocampus, in contrast, while we observed evidence for representing the current state (0.0263 CI [0.0079, 0.0446]; $t_{(30)} = 2.93$, $p = 0.007$, $t$ test), we did not find any evidence for increasing representation of future states (interaction 0.008 CI [−0.018, 0.035]; $t_{(26)} = 0.64$, $p = 0.53$, $t$ test, Supplementary Fig. 6). This effect was statistically equivalent to a null effect ($t_{(26)} = 2.32$, $p = 0.014$, TOST). The future state interaction effect in the hippocampus was non-significantly weaker than the effect in the OFC-VMPFC ($t_{(25)} = 1.57$; CI [−0.0931, 0.126]; $p = 0.129$, paired $t$ test). While we observed significant representations of the current state in our control regions in the posterior and ventral occipital cortex that responded to the categories of visual stimuli (0.318 CI [0.292, 0.345]; $t_{(30)} = 24.62$, $p < 0.001$, $t$ test), we did not find evidence for increasing representation of future states (interaction −0.005 CI [−0.0213, 0.012]; $t_{(30)} = -0.5722$, $p = 0.57$, $t$ test; Supplementary Fig. 6). This effect was statistically equivalent to a null effect ($t_{(30)} = 2.38$, $p = 0.012$, TOST). The future state interaction effect in the visual cortex was also significantly weaker than the effect in the OFC-VMPFC ($t_{(28)} = 2.76$; CI [0.014, 0.098]; $p = 0.01$, paired $t$ test).

**Multivariate control analyses**. We conducted several additional analyses to support the finding of future state representation across learning in the OFC-VMPFC. First, in a control analysis modeling each correct repetition bin separately, we found no pattern of change over time in current (state 1) representation in the OFC-VMPFC (Supplementary Fig. 4). As expected, we observed an increase in future (state 3) information over time (Fig. 3c) such that the strength of representation of the current visible state and the state anticipated >30 s in the future was no longer significantly different by the last learning repetition ($t_{(28)} = 1.91$; CI [−0.005, 0.152]; $p = 0.066$, $t$ test). We also confirmed that the increase in future state information in the OFC-VMPFC across learning was not sensitive to our exclusion of two participants who did not meet the orthogonal requirement of above-zero classification of both state 1 and state 3 on-screen stimuli. Across the full 31 participants, we continued to find a significant future state by correct repetition interaction (0.046 CI [0.006, 0.086]; $t_{(30)} = 2.33$, $p = 0.027$, $t$ test; Supplementary Fig. 3). Next, in two control regression models, we removed the current state or the future state from the model and found similar results for the included variables as in the full model above (Supplementary Fig. 4), indicating that the result was not due to unexpected interactions between included variables. We then examined whether the pattern of future state representation was found across all three stimulus categories that made up the combined analysis or whether it was driven by particular categories. Indeed, we found a positive effect of the future state by correct repetition interaction across the separate face, scene, and object analyses that did not differ between categories (Supplementary Fig. 5).

In a final control analysis, we examined the specificity of the OFC-VMPFC future by correct repetition effect. We examined 25

different ROIs covering the PFC based on functional coactivation patterns across published neuroimaging studies[38] (Supplementary Fig. 10) plus four additional hippocampal subregions. Across all of the PFC subregions, one lateralized sub-part of the left dorsolateral PFC exhibited an effect at the 5% level ($p < 0.0051$, $t$ test; Supplementary Table 2). Given the large number of exploratory regions considered and the lack of multiple comparisons correction, we do not interpret this result further. In the hippocampus, as an additional check of specificity, we further examined whether significant representations of future states were present but hidden by using the bilateral ROI. We separately analyzed the right anterior, left anterior, right posterior, and left posterior hippocampus; across these four additional hippocampal ROIs, we found no future by correct repetition interaction effects ($p$-values >0.40, $t$ test; Supplementary Table 2). The lack of general effects supports our a priori focus on the OFC-VMPFC, indicating relative selectivity. In turn, this selectivity indicates that our results are not likely to be due to a general confound in the design or analysis that would frequently produce a false-positive result.

We next examined whether information representation at subsequent states in the maze trials reflected future expectations or recent experience. At state 2, which follows state 1 after ~7 s but does not include any stimuli on the screen, we found a positive but non-significant change in the representation of the future state 3 across learning in the OFC-VMPFC (interaction 0.040 CI [−0.002, 0.082]; $t_{(28)} = 1.95$, $p = 0.062$, $t$ test; Supplementary Fig. 7). This state 2 interaction was in the same direction as the effect at state 1, potentially indicating that OFC-VMPFC representations of future states across learning continued through state 2. We found no interaction effects in the hippocampus or visual control regions (Supplementary Fig. 7).

At state 3, we did not find any evidence for the representation of preceding states (information reflecting state 1 during state 3) in the OFC-VMPFC (Supplementary Fig. 8). However, we did find that information about the current stimulus decreased across learning (interaction −0.060 CI [−0.108, −0.012]; $t_{(28)} = -2.56$, $p = 0.016$, $t$ test; Supplementary Fig. 8), which could reflect suppression of responses to presented stimuli as they become increasingly predicted by this circuit. A similar negative effect was observed in the visual control regions (−0.023 CI [−0.040, −0.007]; $t_{(28)} = -2.95$, $p = 0.006$, $t$ test). Finally, during reward feedback, we found no significant representation of past states in any ROI (Supplementary Fig. 9).

**fMRI connectivity results**. We finally returned to univariate analyses to examine whether the OFC-VMPFC and hippocampus may interact. In an initial general PPI analysis using the OFC-VMPFC region as a seed, we found robust positive connectivity between these two regions across different maze states, including state 1, the feedback state, and rest ($p < 0.0001$ SVC; Supplementary Fig. 11). A conjunction of results across these three states showed that in addition to the hippocampus, activity in only a few clusters including the posterior cingulate and dorsomedial PFC remained at stringent uncorrected thresholds.

In a PPI model specifically testing for learning effects, we examined changes in OFC-VMPFC connectivity with the rest of the brain across repetitions of mazes that were successfully learned. We found no significant positive or negative main effects of learning repetition on OFC-VMPFC connectivity. However, we found a significant positive relationship between the strength of OFC-VMPFC connectivity and individual differences in the OFC-VMPFC acquisition of future state representation in a cluster including the right hippocampus, midbrain, and thalamus (Supplementary Fig. 12; Supplementary Table 3). This indicates

that increases in the OFC-VMPFC representation of future states across learning were related to concurrent increases in functional connectivity between the OFC-VMPFC and hippocampus. Additional clusters included the putamen as well as the dorsal medial and lateral PFC (Supplementary Fig. 12; Supplementary Table 3). As a control, we conducted the same PPI analysis on later states in a maze trial (state 2, state 3, and the feedback state); here we found no significant relationship between OFC-VMPFC connectivity and the OFC-VMPFC future state interaction effect (Supplementary Table 3).

## Discussion

Learning to predict the future is a critical aspect of goal-directed decision-making. While previous research has focused on the relatively short-timescale prediction of sensory and motor events[1,4,5], we do not yet know how the brain acquires predictive representations of distal future states during goal-directed learning. In a simple learning task with extended sequences of experience, we found that participants learned over only a few repetitions to make correct choices. Using fMRI, we demonstrated that this rapid learning across repetitions—each separated by minutes—was related to initially high and then decreasing activity in the hippocampus and parahippocampal cortex. Critically, as learning progressed, we found an increasing representation of to-be-visited future states in multivariate patterns of activity in the OFC-VMPFC. By the last learning repetition, information representing a hidden state ~30 s in the future was of similar strength as the state currently visible on the screen. Our results indicate that a learning mechanism in the OFC-VMPFC may enable the human brain to represent the distant and unseen future consequences of goal-directed actions.

Our finding that the human OFC-VMPFC acquires distal future state representations provides novel support for the proposal that the OFC represents a cognitive map of the environment[6,15–17]. Computationally, the representation of future state information in the OFC-VMPFC also lends support to reinforcement learning models that explicitly represent state-to-state associations, such as model-based or successor representation models[24–26] over additions to model-free temporal difference algorithms, such as eligibility traces[21]. However, we cannot rule out a contribution of a cooperative model-free eligibility-trace learning mechanism that could assist in state and action value learning by maintaining preceding state information until feedback[39,40]. While our experimental paradigm was designed to be relatively simple in order to isolate effects of interest during initial learning, our OFC-VMPFC findings have the potential to inform studies of more complex multi-state problems or decision trees[27–29,41], especially in problems where the delay between a choice at one state and an outcome in a distal state is relatively long.

Previous experimental work in humans has shown that recent past (hidden) state information that is necessary for task performance is represented in the medial OFC[15]. In contexts that allow for predictions, other work has also revealed that the OFC may represent information about the identity of immediately upcoming rewards[42,43]. Our results indicate that the medial OFC/VMPFC acquires predictive information over time, allowing the OFC to represent critical future state information for goal-directed decision-making. The >40 s duration of mazes in our experiment, while relatively short in comparison with many experiences outside the lab, also goes well beyond the timescale of common human fMRI studies of decision-making. Other studies have found evidence for visual cortex reactivation for relatively short-timescale predictive associations[12–14], including during reward-based decision-making[14]. However, when state predictions extend beyond several seconds, predictive knowledge must

be supported by a learning system outside of the sensory cortex[5]. Our results indicate that a mechanism in the OFC-VMPFC is capable of fulfilling this role.

In the hippocampus, we found that initially high activity rapidly decreased across learning repetitions. This pattern is consistent with a rapid episodic encoding mechanism, where activity decreases as repetitions contain less new information[44]. Notably, in the current experiment, repetitions of mazes were separated by at least 4 min, a delay that likely requires a contribution of longer-term memory for successful learning. Nevertheless, we also replicated the finding of a role for working memory in the maintenance of recently learned value associations[30]. Our previous study used learning from immediate feedback to examine the distinct but related question of spacing of learning sessions across days instead of minutes. While the vast majority of previous human studies have examined memory for brief episodes (e.g., static pictures), our results extend what is known about learning-related hippocampal effects by showing that hippocampal activity decreases across learning selectively at the beginning of goal-directed episodes. This finding, however, contrasts with a previous report which found increases in event-onset hippocampal activity over repeated viewing of passive movies[45]. These different effects could be related to the goal-directed learning component of this study, where participants are incentivized to predict future consequences. Behaviorally, the rapid learning we observed after even a single repetition fits well with recent research showing that humans can make value-based decisions based on information provided by specific past experiences[46–48]. For goal-directed decision-making, this rapid encoding of maze experiences may work together with other learning mechanisms, potentially in the OFC, to allow us to seek out positive outcomes or avoid negative outcomes.

While we found that the OFC-VMPFC represented distal future states, we did not find evidence for future state representation in the hippocampus. Such a null effect is surprising, as research in rodents has demonstrated that the hippocampus represents future paths[10,49]. However, we did find a significant correlation between the increase in functional connectivity between the OFC-VMPFC and hippocampus across learning and the increase in OFC-VMPFC representation of future states across learning. Previous research in humans has found that hippocampal activity during reward learning co-varies with the activation of associated items in the visual cortex[12], and that hippocampal activity patterns represent retrieved information about currently cued target locations[50]. Recent papers have reported reactivation of hippocampal activity patterns during rest periods following learning[51,52] and that hippocampal activity co-varies with the reactivation of visual stimuli[53]. Interestingly, in recent work using MEG, an imaging method with much faster temporal resolution, Kurth-Nelson et al.[54] reported that fast sequences of activity reflected potential trajectories in a non-spatial environment. While no correlations between MEG activity and behavior were identified in that study, such rapid sequence-related activity, likely decoded primarily from the visual cortex, could be driven by the hippocampus in humans, similar to findings in rodents[10,11].

One limitation of the experiment was that our focus on relatively long timescales of future representation limited the design to include only eight unique maze structures. For comparison, Schuck et al. report a stronger effect when examining OFC representation of the relevant task context (indicated by the previous and current trial stimuli) in a rapid event-related design with more than ten times as many trials. Importantly, our OFC-VMPFC result is supported by numerous additional analyses that indicate specificity in time and consistency across different categories. The effect was also selective to the OFC-VMPFC, such that at an uncorrected level only 1 of 25 additional prefrontal regions of interest exhibited

an effect. However, it will be important to replicate these findings in future studies and related paradigms.

The brain has been proposed to be a predictive machine[1–3]. Our results provide new support for how the brain is able to make relatively long-timescale predictions that are critical for goal-directed decision-making. We find that as models of the environment are initially established, a learning mechanism in the OFC-VMPFC is capable of making long-timescale predictions about future states. At the same time, overall activity in the hippocampus correlated with maze exposure across learning. Our complementary results in the OFC-VMPFC and hippocampus, combined with their connectivity relationship, suggest that these regions may interact to support learning and decision-making in goal-directed tasks[19,20]. Further understanding of these mechanisms and their interaction will be important for understanding deficits in both learning the structure of the environment and in using these representations for goal-directed decisions that are found in various psychiatric disorders and addiction.

## Methods

**Participants**. A total of 38 subjects participated in the experiment. Participants were recruited via a list of prior participants in MRI studies who had consented for re-contact, postings on an online university bulletin board, and referral from other participants. Participants were fluent German speakers with normal or corrected-to-normal vision and no self-reported neurological or psychiatric disorders or depression as indicated by questionnaire. Three participants were excluded based on behavior: one for previous participation in a related experiment, one for sleeping during the task, and one for chance-level performance. Thus, data from 35 participants were included in the behavioral analysis (21 female; mean age, 25.7 years; range, 19–31 years). After additional fMRI-based exclusion (detailed below), data from 32 remaining participants (18 female) were included in the univariate fMRI analysis. One additional participant was excluded from the multivariate analysis due to an error in the localizer phase. All participants were remunerated for their participation, and participants were additionally compensated based on rewards received during maze learning. The Ethics committee of the Medical Chamber Hamburg approved the study and all participants gave written consent.

**Learning phase**. The navigation learning task consisted of a learning phase followed by a localizer phase (Fig. 1a, c). In the learning phase, across four repetitions for each of eight Y-maze sequences (plus two final non-analyzed mazes), participants attempted to learn the correct initial choice at state 1 in order to progress through the correct state 3 and reach a reward at the terminal feedback state. Maze- and state-unique stimuli from three categories (faces, scenes, and objects) were presented at state 1 and state 3 to enable decoding of state representations across learning. For the sequence design, we utilized an adapted Y-maze structure with a choice at state 1 and instructed choices at states 2 and 3 to both allow for rapid learning (by limiting the structure to two branches) and relatively long maze durations without excessive scan duration.

In each maze, the participant navigated through an illustrated series of rooms (states) with hallways in-between states (Fig. 1a, c). At each stage in a maze, focal stimuli were presented in the center wall of the room. Two walls with space for doors extended to the left and right of the center wall. In state 1 and state 3, a maze- and state-unique stimulus was presented on the center wall. In state 2, a gray square was presented on the central wall. In state 1 and 3, both the left and right door options were initially visible. In state 2, only the single available pseudo-randomly determined (and fixed throughout learning) left or right door option was visible. In state 3, the non-available door was hidden after a short duration. In the reward state, no door options were visible. Between states, an illustrated hallway was presented with an occasional central upward-pointing arrow indicating to participants to press a key to continue advancing to the next state.

Maze duration, from state 1 until the end of the feedback state, was on average 48.5 s for the eight mazes of interest (range, 38.3–77.4 s). The mean time between state 1 onset and reward feedback indicating the accuracy of the state 1 choice was 42.8 s (range, 32.9–71.4 s). This delay from stimulus and action to feedback is well past the short range of time where dopamine responses show fidelity to a reward-prediction error signal[22,23]. The mean time between state 1 onset and state 3 onset was 31.2 s (range, 22.1–58.9 s). For the first repetition, total maze duration was ~10 s shorter due to abbreviated inter-state periods (36.7 s mean duration), leading to shorter durations between state 1 and state 3 or reward. The delays between repetitions of the same maze were on average more than 4 min. Specifically, the delay between repetition 1 and repetition 2 was 4.4 min (mean 266.0 s, range, 104.4–430.2). The delay between repetition 2 and repetition 3 was 4.6 min from (mean 278.5 s, range, 102.2–1294.6). The delay between repetition 3 and repetition 4 was 6.9 min (mean 414.0 s, range, 250.5–1266.0). This large delay strongly reduces the likelihood

of between-repetition working memory maintenance as an explanation of learning performance[30,31], a common problem in reward learning paradigms where learning repetitions are separated only by several seconds on average.

At state 1, an illustration of a room with a superimposed central stimulus was presented for 4 s, while a small white fixation cross was shown over the stimulus. When the fixation cross disappeared, participants were allowed to make their choice. After making a choice, the selected door opened slightly and the central stimulus moved to the selected door for 1.5 s. Next, the screen changed to a hallway phase for the inter-state period (described below). In states 2 and 3, the room illustration and central stimulus were shown for 2 s (state 2) or 4 s (state 3), after which the stimulus moved to the computer-determined (and still closed) door. Next, participants selected the indicated correct left or right button (for a maximum 30 s duration). A warning appeared if the participant was too late or if they selected the incorrect door. The selected door then opened slightly and the central stimulus moved to the door for 1.5 s. The screen then changed to the hallway view. When participants reached the feedback state, the center wall was blank. They were required to press the bottom button to reveal the feedback. If the correct choice was made (or for the first repetition, if the maze had been assigned to be rewarded), a euro coin was displayed. Alternatively, a 50-cent euro coin with an overlaid red "X" was displayed to indicate loss feedback. The feedback image was shown for 5 s. During the inter-state period between all states, after a 0.25 s blank screen, an illustration of a hallway was presented. An upward-pointing arrow appeared after 0.25 s. Participants progressed to the next state by pressing the top button when the arrow appeared. A series of two arrows, separated by a short delay, followed the first and second state, while one arrow followed the third state. The average duration from start to the end of a maze experience was 48.5 s. Trials were followed by a mean 16 s (range, 15–18.5) inter-trial-interval during which a central white fixation cross was presented; the fixation cross changed to black for the 1 s preceding the start of the next maze.

For the eight mazes of interest, reward feedback was given on the first exposure for four mazes (independent of the state 1 left or right choice of the participant), and miss feedback was given on the first repetition for the other four mazes. Based on the initial state 1 left or right choice of the participant and whether the maze was assigned to end in an initial reward or loss, the "correct" and "incorrect" choice was assigned for subsequent repetitions for each maze. The mazes were presented in a pseudo-random order with staged introduction of new mazes as follows. The first four mazes were shown through three repetitions, such that each maze was repeated one time before another maze could be repeated again. Intermixed with the fourth repetition, two mazes from the second set were interleaved. At the fourth repetition for the second set of mazes, two additional mazes were interleaved. These additional mazes always ended in a loss in order to better balance the reward and loss associations for probe questions in the subsequent localizer. During scanning, the experiment was divided into four blocks of ten maze trials per block.

To measure internal activation of future state representations, each maze in the experiment included three categories of stimuli: faces, scenes, and objects. These were assigned to state 1 and the two alternative state 3 states. Color pictures of female faces set on a gray background were drawn from the Stanford Face Database, while scenes were drawn from an internal database; both sets of stimuli have previously been utilized to identify internal reactivation of state representations[12]. Color pictures of objects were drawn from a previously used set of images compiled via internet search[55], composed of medium-sized items that can be manually interacted with (e.g., a book, water bottle, or guitar) set on a white background. Three pseudo-random orderings of the maze- and state-unique picture stimuli were used for counterbalancing. To indicate reward or loss feedback, a picture of a 1 euro coin was used for reward feedback while a picture of a 50-cent euro coin with a red "X" drawn through it was used for miss feedback. In addition, for reward feedback, outwardly moving gold-colored dots were shown behind the coin to allow us to examine a potential motion-sensitive cortical response to rewards and anticipated rewards. To achieve this effect, a rapidly presented sequence of 12 static images started with only a few visible dots around the coin and then progressed as these and additional dots moved outward, giving the impression of a burst of dots moving away from the coin. These 12 images were repeated in sequence three times during the 5-s feedback period.

Before entering the MRI scanner, participants completed 1 trial of a practice maze. This practice maze was repeated again inside the MRI before the start of the experiment to ensure participants were familiarized with the screen and button box.

**Localizer phase**. To measure patterns of activity associated with the stimulus categories seen at states 1 and 3 in the learning phase, we next collected a "localizer" phase in the MRI scanner. During each trial, participants viewed either a stimulus from the learning phase, a scrambled object stimulus, or moving dot stimulus. For the face, scene, and object stimuli from the learning phase, participants were instructed to try to remember whether that stimulus was followed by reward or loss during learning so that they could perform well on occasional probe questions.

The localizer phase included 155 total trials, composed of 120 face, scene, and object trials, 20 scrambled object trials, and 15 "motion" trials with a moving dot background. The motion stimuli were composed of a gray hexagon with black outline and blue dots shifting across 12 static images, modeled after the reward

feedback background gold dots seen during the learning phase. On each trial, static stimuli were presented for 2 s. For motion trials, the stimulus duration was 3 s. If a memory probe followed the picture, the text "Reward? ^" and "Loss? ˇ" appeared above and below the stimulus picture, respectively. Participants indicated their response with the corresponding top and bottom buttons. After a 0.5-s delay followed by a 0.25-s fixation period, a confidence rating scale appeared along with the text "How certain?". The rating scale was composed of four levels of confidence: "not at all", "a little", "medium", and "very", which participants navigated using the left and right buttons on the box, confirming their answer with the bottom button. After a 0.25 s pause, the task continued with the inter-trial-interval indicated by a fixation cross (mean 2.5 s duration; range, 1.5–4.5 s).

The localizer phase was composed of three mini-blocks, each containing 40 face, scene, and object stimuli, 4–8 scrambled object stimuli, and 5 motion stimuli. Face, scene, and object trials were pseudo-randomly ordered, while scrambled object stimuli were shown at the end of the mini-block in two sets of four, preceding and following a set of five motion stimulus trials. Probe questions that assessed memory for the reward or loss associated with that stimulus during learning were included after 25% of the face, scene, and object trials to increase participant attention and to assess memory for the reward or loss association. The base list was modified for each participant, such that stimuli that were not seen during the learning phase were replaced with an additional 2 s inter-trial-interval (mean, 9.9; range, 3–12). One participant was excluded from localizer analyses and multivariate analyses because an error in list creation led to an exclusion of 57 trials, yielding too few trials for accurate category classification.

After exiting the MRI scanner, participants completed two additional measures. We first collected the Beck Depression Inventory (BDI). The second measure we collected was the operation-span task (OSPAN), which was used as an index of working memory capacity[56,57]. In the OSPAN, participants made accuracy judgments about simple arithmetic equations (e.g., "2 + 2 = 5"). After a response, an unrelated letter appeared (e.g., "B"), followed by the next equation. After arithmetic-letter sequences ranging in length from 4 to 8, participants were asked to type in the letters that they had seen in order, with no time limit. Each sequence length was repeated 3 times. In order to ensure that participants were fully practiced in the task before it began, the task was described with in-depth instruction slides, followed by five practice trials. Scores were calculated by summing the number of letters in fully correct letter responses across all 15 trials[30,57].

The experimental tasks were presented using Matlab (Mathworks, Natick, MA) and the Psychophysics Toolbox[58]. The task was projected onto a mirror above the participant's eyes. Responses during fMRI scanning were made using a 4-button interface with a "diamond" arrangement of buttons (Current Designs, Philadelphia, PA). At the end of the experiment, participants completed a paper questionnaire querying their knowledge of the task instructions and learning strategy. For all parts of the experiment, verbal and on-screen instructions were presented in German; for the methods description and task figures, this text has been translated into English.

**fMRI data acquisition**. Whole-brain imaging was conducted on a Siemens Trio 3 Tesla system equipped with a 32-channel head coil (Siemens, Erlangen, Germany). fMRI measurements were performed using single-shot echo-planar imaging with parallel imaging (GRAPPA, in-plane acceleration factor 2; TR = 1240 ms, TE = 26 ms, flip angle = 60; 2 × 2 × 2 mm voxel size; 40 axial slices with a 1 -mm gap)[59] and simultaneous multi-slice acquisitions ("multiband", slice acceleration factor 2)[60–62] as described in ref. [63]. The corresponding image reconstruction algorithm was provided by the University of Minnesota Center for Magnetic Resonance Research. Slices were tilted ~30° relative to the AC–PC line to improve signal-to-noise ratio in the orbitofrontal cortex[64]. Head padding was used to minimize head motion. Three participants were excluded for excessive head motion (a total of 5 or more >2.0 mm framewise displacement translations from TR to TR per learning-phase block).

During the learning phase, four functional runs of an average of ~10 min and 45 s were collected, each including ten maze trials. During the localizer phase, one functional run of an average of ~15 min was collected. For each functional scanning run, four discarded volumes were collected prior to the first trial to allow for magnetic field equilibration and to collect reference data for multiband reconstruction.

Structural images were collected using a high-resolution T1-weighted magnetization prepared rapid acquisition gradient echo (MPRAGE) pulse sequence (1 × 1 × 1 mm voxel size) between the learning phase and the test phase.

**Behavioral analysis**. Behavioral analyses were conducted in Matlab 2016a (The MathWorks, Inc., Natick, MA). The results presented below are from the following analyses: t tests vs. chance for learning performance, within-group (paired) t tests comparing differences in reward- and loss-associated stimuli across conditions, Pearson correlations, and Fisher z-transformations of correlation values. We additionally tested whether non-significant results were weaker than a moderate effect size using the Two One-Sided Test (TOST) procedure[37,65], as implemented in the TOSTER library in R[37]. We used bounds of Cohen's d = 0.53, where power to detect an effect in the included group of 32 participants was estimated to be 80% (d was adjusted accordingly when analyzing subsets of participants).

**fMRI data analysis**. Preprocessing was conducting using FMRIPREP version 1.0.0-rc2 (http://fmriprep.readthedocs.io) on openneuro.org, which is based on Nipype[66]. Slice timing correction was disabled due to short TR of the input data. Each T1-weighted volume was corrected for bias field using N4BiasFieldCorrection v2.1.0[67], skullstripped using antsBrainExtraction.sh v2.1.0 (using the OASIS template), and coregistered to skullstripped ICBM 152 Nonlinear Asymmetrical template version 2009c[68] using nonlinear transformation implemented in ANTs v2.1.0[69]. Cortical surface was estimated using FreeSurfer v6.0.0[70].

Functional data for each run was motion corrected using MCFLIRT v5.0.9[71]. Distortion correction for most participants was performed using an implementation of the TOPUP technique[72] using 3dQwarp v16.2.07 distributed as part of AFNI[73]. Functional data were coregistered to the corresponding T1-weighted volume using boundary-based registration with 9 ° of freedom implemented in FreeSurfer v6.0.0[74]. Motion correcting transformations, T1-weighted transformation, and MNI template warp were applied in a single step using antsApplyTransformations v2.1.0 with Lanczos interpolation. Framewise displacement[75] was calculated for each functional run using Nipype implementation. For more details of the pipeline see http://fmriprep.readthedocs.io/en/1.0.0-rc2/workflows.html. The data were not resampled; voxel size remained at 2 × 2 × 3. For univariate analyses, images were then smoothed with a 6 mm FWHM Gaussian kernel.

General linear model analyses were conducted using SPM (SPM12; Wellcome Trust Centre for Neuroimaging). Prior to modeling, we applied a dilated mean anatomical mask to the functional data in order to provide a reasonable constraint on whole-brain multiple comparisons correction at the second level. The data from the four learning-phase blocks were then concatenated, using a method developed by T. Sommer and adapted from Schultz et al.[76]. An autoregressive model, AR(1) (adapted to account for the concatenated sessions), was used to model serial autocorrelations in the data. fMRI model regressors were convolved with the canonical hemodynamic response function and entered into a general linear model (GLM) of each participant's fMRI data. Six scan-to-scan motion parameters (x, y, z dimensions as well as roll, pitch, and yaw) produced during realignment were included as additional regressors in the GLM to account for residual effects of participant movement.

A univariate GLM was constructed to examine BOLD correlates of learning across repetitions and responses to reward feedback. We predicted that brain regions involved in encoding the stimuli and the associations in the maze episode would show a rapid decrease (or decay) in activity across repetitions. To capture this decrease, we created a decaying encoding predictor that decreased exponentially across repetitions. Specifically, a geometric decay of 50% was applied across the four repetitions, with values for the four repetitions of: 1, 0.50, 0.25, and 0.125. This encoding decay regressor was entered as a separate parametric modulator of activity at states 1, 2, and 3, although our primary prediction was about activity at the start, in state 1. At the reward state, reward and loss feedback were modeled as 1 and −1. We additionally modeled reward feedback during the subsequent rest period to test for lasting effects of reward on BOLD. States 1 and 3 and the feedback state were modeled as 5 s duration events. State 2 was modeled as a 4 s duration event. The rest period was modeled as a 12 s duration event. The 5 s duration of the State 1 and 3 primary events of interest captures the 4 s presentation of the central stimulus plus 1 additional second where the stimulus had shifted to the left or right door and matches the duration of the modeled feedback state events. Responses to the eight mazes of interest in addition to the first exposure to maze nine and ten were included in the model. (Later repetitions of maze nine and ten were excluded because both choices were predetermined to lead to a loss.)

Our navigation learning task, where reward associations remained fixed across repetitions, did not allow for us to examine effects of choice value at state 1 or feedback due to collinearity with learning and reward variables. As a consequence, we could not examine whether responses to delayed reward feedback were better captured by reward magnitude versus reward prediction error, as suggested by neurophysiological studies in non-human primates[22,23]. Specifically, reward and choice value, the components of reward prediction error, were highly positively correlated (median r = 0.57) and thus we could not include them together in a model at the same time period. At stage 1, decaying encoding across repetitions and choice value were also highly negative correlated (median r = −0.71) and could not be included together in a model. At feedback, while the correlation between decaying encoding and reward was lower than the above correlations (median r = −0.46), we opted to not model decaying encoding at feedback or post-feedback rest.

Next, we analyzed the localizer phase to identify univariate patterns of activity that discriminated between the stimulus categories. These contrasts were used to identify regions of interest for multivariate analyses. The initial GLM included separate regressors for faces, scenes, objects, scrambled objects, and motion stimuli. Static localizer stimuli were modeled as 2 s events while motion stimuli were modeled as 3 s events. From these first-level results, we computed contrasts with each of the three categories compared versus either of the two alternatives as well as each category versus the combination of both alternatives.

We additionally conducted psychophysiological interaction (PPI) analyses to examine functional connectivity between the OFC-VMPFC and the hippocampus. A general PPI was estimated first, looking at main effects of connectivity at different states in the maze trials. Each state was modeled as a 6 s event, and rest

was modeled as a 12 s event. The OFC-VMPFC ROI was as described above. Three general PPIs were estimated for state 1, the feedback state, and rest state. We computed a conjunction of these results to examine common OFC-VMPFC connectivity across multiple states during maze navigation.

A second PPI specifically examined changes in connectivity at state 1 across repetitions for successfully learned mazes. The psychological term in the PPI was a contrast between repetitions 2 and 3 (late learning) and repetitions 0 and 1 (early learning). Mazes where above-chance performance was not achieved were omitted from this contrast. Our goal in this analysis was to examine changes in connectivity that may relate to learning, so we additionally entered the OFC-VMPFC future by correct repetition effect as a covariate. This analysis tested for increases in OFC-VMPFC connectivity across learning that positively related to increases in OFC-VMPFC future state representation across learning. Additional PPI models were estimated for state 2, state 3, and the feedback state as controls.

**Multivariate fMRI analyses**. Our tests of predictive representations relied on multivariate analyses because univariate analyses are not able to analyze representation content across multiple categories in a single region. In the learning phase, we estimated a mass-univariate GLM using the non-smoothed BOLD data. Each event in each stage as well as the rest period were modeled with individual regressors, yielding 200 regressors of interest for the learning phase. The learning-phase event durations were as above: 5 s for states 1 and 3 and the reward state, 4 s for state 2, and 12 s for the rest period. In the localizer phase, we estimated a separate mass-univariate GLM using the non-smoothed BOLD data. Each trial was modeled as a 2 s event with an individual regressor, yielding 155 total regressors. Models included six motion regressors of no interest. In the learning phase, block regressors were also included as effects of no interest.

In general, multivariate analyses were necessary to test our hypotheses for several reasons, including the following: when focusing on a single region, univariate analyses are insensitive to distributed patterns of information in cases where average signal does not differ between stimulus categories. Moreover, univariate analyses of a single region are not able to uniquely disambiguate between potential explanations for changes in responses over time, as any change could be due to multiple factors including attention, changes in the processing of the on-screen stimulus, and increases or decreases in response to one distal state option or the other distal state option.

Multivariate analyses were conducting using The Decoding Toolbox[77]. Classification utilized a L2-norm learning support vector machine LIBSVM[78] with a fixed cost of $c = 1$. The classifier was trained on the full localizer phase data, which was divided into three mini-blocks. In the case of imbalanced numbers of face, scene, and object trials per mini-block (due to omitted stimuli not seen during learning), other trials were randomly left out to achieve balance. The trained classifier was then tested on the full learning-phase data. Note that for the primary across-phase classification analysis, no cross-validation is necessary for training because no inferences are drawn and no results are reported on the localizer phase data. Based on the strength of discriminability in the localizer phase (using cross-validation), classification of faces was based on the comparison of faces versus scenes. Classification of scenes was based on the comparison of scenes versus faces. Classification of objects was based on the comparison of objects versus faces. While scrambled objects were included in the localizer phase to compare to objects, classification performance for objects versus scrambled objects was worse than that for objects versus faces. For objects in general, across OFC-VMPFC, hippocampus, and visual ROIs, classification performance was lower than that for faces or scenes. Learning-phase classification is reported as area under the curve (AUC) which uses graded decision values and better accounts for biases in classification that may arise due to the different processes engaged by the localizer and learning phases.

The representation of current and future state information was analyzed by applying classifiers trained on the localizer phase to the patterns of activity evoked during maze learning, with a focus on the state 1 onset of each maze repetition. As described above, one classifier for each category was trained based on the localizer phase data. These trained classifiers were applied to each stage of each maze trial to derive decision values, a parametric variable ranging from 0 to 1 representing how well the pattern of multivoxel activity in regions of interest at each state was described by the classifier. Importantly, because the classifier was trained after learning and all three categories would end with approximately the same average positive expected reward value, discrimination of localizer patterns would not be systematically influenced by learned value. Given this balance of value in the training data, it is not possible for any changes in classification over learning to be affected by shifts in value alone across learning.

The decision values for each classifier and for each state were then entered into regression models to measure representations of current and future states. We computed three regression models, one for each category, to examine the information present in classifier decision values at state 1. These models estimated whether decision values were related to state 1 category, state 3 category, the number of correct maze repetitions (derived from behavior), and the two interactions between state 1 and 3 category and the number of correct repetitions. State 1 and state 3 category were represented as binary variables, where 1 indicated that the state matched the category of interest and 0 otherwise. The number of correct repetitions variable was based on learning performance for each maze, focusing on whether the state 3 stimulus of interest had been reached during

learning. For the first repetition, the value was set to 0. The value remained at 0 until the repetition following when the participant reached the correct state 3 and was able to observe the stimulus (and category) present in that state. Subsequently, the value incremented by 1 for every subsequent repetition. We made no assumptions about whether future predictions would be stronger or weaker if participants returned again to the incorrect choice. An alternative model where the correct repetition variable was only incremented on correct trials yielded qualitatively similar results. It is important to note that the use of a regression analysis across decision values for all trials provides a regression coefficient which we compare to a null (zero). However, this analysis does not yield a percent correct measure such as those reported in many classification studies, and consequently the resulting values should not be interpreted as a percentage measure. Note that while the localizer followed learning, any significant effects of learning on future state representation cannot be due to the classifier detecting activation of learned associations at test: any such confounding effects would only affect the main effect of future state representation, not the change of this representation across learning.

Within-participant, the results from the above regression models were averaged across the three categories to allow for group-level analyses (via $t$ tests). This approach was used instead of a single hierarchical mixed-effects model because each trial would be included three times in a single model, violating independence of observations. Participants' data for individual categories were excluded from averaging across models if the classifier failed to generalize across task phases and classify learning-phase state 1 and state 3 on-screen stimuli. Performance was calculated as the regression coefficient between (concatenated) decision values for state 1 and state 3 and the contrast between the presence of the actual category of interest (e.g., faces) versus the comparison category (e.g., scenes) at state 1 and state 3. For two participants, the classifiers failed to generalize to any category during the learning phase and data from these participants were excluded from further analyses.

It has been shown that it is not valid to conduct statistical inference specifically on cross-validated classification accuracy measures of information using $t$ tests[36]. In part, as informational measures cannot be below zero, assumptions underlying the $t$ test are violated for cross-validation within the same dataset. Our training and testing was conducted on separate datasets ("cross-classification" between the localizer and the learning phase) which does allow for potential "true" below-zero values, a case not addressed by Allefeld et al.[36]. Furthermore, we found that our cross-classification results for current and future state in the OFC-VMPFC follows a normal distribution (Anderson–Darling goodness-of-fit hypothesis test). While the above concern may still apply to inferences made about the main effects of current and future state in our regression approach, our primary hypothesis rests on the change in information about future states across learning repetitions and not on comparing whether information content is different than zero. Thus, the use of a $t$ test is valid.

A first control analysis examined whether the exclusion of two outliers affected the primary future state by correct repetition results. Other analyses examined current, future, and past state representations at different states in the task (state 2, state 3, and the feedback state). We did not explore timepoint-by-timepoint modeling of the rest period due to signal-to-noise concerns: the category decoding of current states was at a relatively low level and the search through many timepoints of data was less constrained than the modeling of specific states as above.

Additional control analyses examined different versions of the primary model above. First, we examined four separate models, one for each correct repetition bin, with current state and future state as predictors. This allowed us to examine the pattern of change in current and future state representation over time without the interaction included in the primary model. Second, we examined two alternative models where either current state or future state was omitted from the model. This allowed us to examine whether the inclusion of both current and future state in the same model significantly affected the results. Next, we examined the generality of the future state by repetition effect: as the primary results collapse across the three stimulus categories (faces, scenes, and objects), we examined whether results were driven by a particular category or whether the same pattern was consistently observed across categories.

Finally, we examined whether other parts of the PFC represented information about future states across learning. This analysis informs whether any pattern of effects found in the OFC-VMPFC, hippocampus, or visual cortex was unique or instead commonly found throughout other frontal cortical regions. A common approach to whole-region analyses is to use a "searchlight" analysis, based on classification accuracy in a spherical region around each voxel. Our primary multi-category and multiple regression analysis of state information representation change (detailed above), however, makes a common searchlight approach unwieldy. As an alternative, we examined regions of the PFC in an approximately tiled manner. This approach has a benefit over searchlight analyses in that it pools voxels together that have a common functional architecture, thus respecting the boundaries of different functional regions.

We conducted the classification and regression analyses separately in the left and right components of the bilateral ROIs, and thus the supplemental ROI analysis included an additional 25 regions (described below). We label the ROIs in Supplementary Table 2, where the abbreviations refer to: anterior OFC (antOFC), perigenual anterior cingulate (pgACC), dorsomedial PFC (dmPFC), frontopolar PFC (fpPFC), lateral PFC (latPFC parts a and b), lateral OFC (latOFC), ACC (ACC parts a, b, and c), subgenual ACC (sgACC), inferior PFC (infPFC), and dorsolateral PFC (dlPFC parts a, b, c, and d). An additional set of analyses investigated the right and left anterior hippocampus (antHipp) and posterior hippocampus (postHipp).

**Regions of interest**. For the univariate results, linear contrasts of SPMs were taken to a group-level (random-effects) analysis. We report results corrected for family-wise error (FWE) due to multiple comparisons[79]. We conduct this correction at the peak level within small volume ROIs for which we had an a priori hypothesis or at the whole-brain cluster level (both after an initial thresholding of $p < 0.005$ uncorrected). For univariate and multivariate analyses, we created two regions of interest in the OFC-VMPFC and hippocampus. We constructed the a priori OFC-VMPFC ROI ($x, y, z$: 0, 44, −14; radius 10 mm) based on a previous finding of short-term preceding state representations in the OFC[15]. The bilateral hippocampus ROI was derived from the Harvard-Oxford atlas—which provided the best overlap with our mean anatomical image—at a threshold of 25%. We focused on the hippocampus instead of the combined hippocampus and parahippocampal cortex to limit the size of the ROI for multivariate analyses. As control regions, we also constructed ROIs based on univariate responses to faces, scenes, and objects in the localizer phase. The face ROI was constructed based on the group contrast of faces versus scenes (derived from smoothed data at the first level) at a threshold of $p < 0.001$, selecting clusters surrounding ($\pm 42 -42 -22$, including $z < -14$ and $y < -68$; $z < 0$ and $y > -68$). The scene ROI was constructed based on the contrast of scenes versus faces at a threshold of $p < 0.001$, selecting clusters encompassing the bilateral parahippocampal gyrus (including $z < 4$ and $y < -68$; $z < 16$ and $y > -68$). The object ROI was constructed based on the contrast of objects versus faces in symmetric clusters in the posterior occipital lobe at a threshold of $p < 0.025$ uncorrected (centered at $\pm 58 -58 -10$).

Our supplemental PFC regions of interest were selected from a 50-region whole-brain parcellation map derived from coactivation patterns across more than 10,000 published studies in the Neurosynth database[38] (http://neurovault.org/images/39711). From this parcellation mask, we extracted 14 ROIs (5 medial and 9 bilateral) in the prefrontal cortex with clustered spatial distributions and sizes approximately matching our OFC-VMPFC ROI, plus one additional bilateral ventral OFC ROI in a region lacking parcellation coverage (Supplementary Fig. 10). The added bilateral ventral OFC ROI was located in the anterior mid-OFC centered at $\pm 22$, 53, −21. A stretched spherical ROI was drawn using a 10-mm radius sphere with centers from $y = 48$ through $y = 54$. The existing ROIs were subtracted from this addition ROI, and the result was masked by the group whole-brain mask to remove voxels clearly outside of the brain. The resulting mask including all the above regions plus our three primary ROIs can be found here: https://neurovault.org/images/122507/. In addition to the PFC ROIs, in this supplemental analysis we also separated the bilateral hippocampal ROI into four parts: right and left anterior hippocampus (defined by $y > -20$) and right and left posterior hippocampus (defined by $y < -20$).

All voxel locations are reported in MNI coordinates, and results are displayed overlaid on the average of all participants' normalized high-resolution structural images using the xjView toolbox (http://www.alivelearn.net/xjview) and AFNI[73].

## Data availability

Behavioral data are available on the Open Science Framework (https://osf.io/gt5kz/). Whole-brain fMRI results are available on NeuroVault (https://neurovault.org/collections/4420/) and the full imaging dataset is available on Openneuro (https://openneuro.org/datasets/ds001019/versions/1.0.0; https://doi.org/10.18112/openneuro.ds001019.v1.0.0).

## Code availability

Code for the multivariate analysis of current and future state representations is available at https://github.com/gewimmer-neuro/ofc-prediction.

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

## Acknowledgements

The authors would like to thank Lara Austerman, Lea Kampermann, and Sven Schönig assistance in behavioral and fMRI data collection and helpful discussions and Martin Hebart and Krzysztof (Chris) J. Gorgolewski for assistance with fMRI methods and preprocessing. The authors are grateful to the University of Minnesota Center for Magnetic Resonance Research for providing the image reconstruction algorithm for the simultaneous multi-slice acquisitions. This work was supported by the Deutsche Forschungsgemeinschaft (Research Fellowship 317946335 to G.E.W.; DFG SFB TRR 58 and SFB 936 to C.B.), and the European Research Council (Grant ERC-2010-AdG_20100407).

## Author contributions

G.E.W. and C.B. conceived the project and designed the experiment. G.E.W. collected and analyzed the behavioral and fMRI data. G.E.W. and C.B. wrote the manuscript.

## Additional information

**Competing interests:** The authors declare no competing interests.

