## [Transparent Peer Review File · Nature Communications]

Reviewers' comments:

Reviewer #1 (Remarks to the Author):

In this paper Wimmer and Buchel investigate the coding of current and predictive state information in a temporally extended Y maze using fMRI. In their task participants move through a Y maze that a number of different states. Crucially, these states include an initial choice state (State 1) and two pre-reward states (State 3) that are dependent on the choice. Each of these three states is represented by a different category of object (faces, scenes or objects) allowing past, present, future (and even counterfactual) states to potentially be decoded using MVPA techniques. Using this approach they find representations of both present and future state in OFC, and representations of present state in hippocampus. Moreover, the representation of future state in OFC gets stronger with learning.

These findings add to mounting evidence that OFC represents state information during learning and decision making tasks. The novelty in this paper is twofold: first they show how *predictive* state information is encoded in OFC over relatively long times (40s); and second they show that these predictive representations develop quickly (after 1 trial, but increasing up to 3 trials). These are important results and, I think, would be appropriate for Nature Comms. Apart from the first point below, my comments are mostly minor.

Perhaps my biggest reservation about this paper is the relatively small sample size (32 after exclusions I believe) that are used to detect relatively small effects. Such small effects are to be expected with a classification paper – especially in OFC – where Schuck et al. also have pretty small effects. This concern is somewhat mitigated by the strong prior hypotheses which, for example, allowed them to define an OFC ROI based on earlier work, but it still puts a question mark on the results for me. If this were a behavioral study I would ask for a replication, for an fMRI study I realize such a request may be too much. However, I would like to see the small effects acknowledged a bit more in the Discussion – perhaps even report a power analysis showing the number of subjects that would be required for a replication of the main findings. Also I think it would be good to compare the classification accuracies with Schuck et al. – are you in the same ballpark for classification accuracy of the current state?

Minor points

As I understand the design of this experiment, because the classifier is trained on a localizer task with no value, there's no confound here between expected value and future state. Is this correct? If so, I think this point should be highlighted, because the first question that comes to mind when reading this is that you are simply decoding value not state.

Would be nice to have a searchlight analysis for classifier. I realize this will be terribly underpowered, but (as with the exploratory PPI analysis) would give a hint of whether there are other regions involved and how specific results are to OFC.

Legends on the supplementary Figures would make for easier reading! Figure S1 especially, but also S3, S4, S5

I see that the data will be shared (thank you!), but will the analysis code also be shared? Also, will the shared data include the excl

Reviewer #2 (Remarks to the Author):

In this paper, Wimmer & Buchel examine the role of the hippocampus and OFC in acquiring predictive information during goal directed learning. Unlike many previous reports, they focus on

rewards/feedback that is extended well beyond just a few seconds. They find that the hippocampus exhibited the highest activity during initial exposure; this hippocampal activation decayed over additional repetitions of the maze. In addition, they also report that patterns within the OFC/VMPFC that represent predictive information about temporally distant states around 30 seconds in the future.

Overall, I think this is a clever study and a well-written manuscript with rigorous analyses. The authors should be commended for the open sharing of data and statistical maps (and presumably analysis code after publication). Nevertheless, I have a few (relatively minor) concerns that should be addressed in a revision.

Primary Issues

1) Perhaps I missed something, but the relationship between states also isn't clear. Understanding how the states (and decisions) are related could put this work into the context of decision trees (e.g., Huys et al., 2012, PLoS Computational Biology; and other work from Peter Dayan). Although these paradigms generally involve feedback at each stage, the overall final outcome is temporally extended, similar to the present work. Thus, I think it would be beneficial to discuss this work and related papers, particularly ones that omit intervening feedback.

2) The description of the task is not completely clear to me. Complicating this matter is the fact that the journal guidelines place the Methods at the end of the paper, making it more important to have a clear task description before presenting the primary results. Thus, I think expanding the figure caption and Introduction to clarify the goals of the maze task at each stage (e.g., what is the subject doing and why) could help.

3) Regarding the predictive OFC responses in Figure 3, are these effects time locked to the initial state? That seems to be the case, but I think it would be important to show that the results are specific to that state and not seen at other time periods and state transitions.

Minor Comments

1) I'm not sure I fully understand the advantage of MVPA in this paper. Could the authors arrive at the exact same conclusions with standard univariate analyses? Is there a specific pattern in the OFC that is predictive? Would that pattern be visible in standard univariate analyses? Please clarify.

2) The neuroimaging results are particularly noisy. Although the authors implement standard corrections for head motion, there's still activation outside the brain and in the white matter (e.g., Figure 2). Is this due to how the task is being modeled or are there potentially sources of noise that are unaccounted for (or both)? I realize there is always unaccounted for variance, but I am concerned that some of the presented results are not within grey matter and could be due to head motion (or other issues).

3) I think it would be helpful for the authors to further clarify the relationship between the motivation for this paper and their recent Journal of Neuroscience paper ("Reward learning over weeks versus minutes increases the neural representation of value in the human brain"). Although the papers seem to be focused on distinct questions, both seem to address the temporal nature and maintenance of learning signals.

4) In Figure 3, what is chance in the predictions? The classification accuracy seems quite low, but the authors don't provide a clear intuition about the nature of chance in these predictions.

Reviewer #3 (Remarks to the Author):

This paper examines whether a representation of a forthcoming state can be decoded from human orbitofrontal cortex when deciding between two alternative courses of action. This question is of interest because of the recent literature suggesting that the hippocampus and orbitofrontal cortex may together represent the structure of the environment, in the form of a cognitive map.

The central result of the paper (figure 3) is that when choosing between two actions at an initial starting state ('state 1'), it is possible to use multivariate pattern analysis on OFC/VMPFC data to: (a) not only decode whether subjects are currently examining a face, house or scene ('current state'); (b) but also whether they will examine a face/house/scene in an intermediate state ('state 3') visited en route to reward, which is modulated by how often that state has been viewed in the past ('future * repetition' interaction).

A strength of the study is that it cleverly examines how a map may be learnt over periods of time that are less susceptible to being solved using working memory (maze repetitions are several minutes apart). It also looks at predictive information over a longer timescale typically than that studied in reinforcement learning tasks. If the claim is robust, it will be of interest to researchers interested in memory, learning and decision making. I also applaud the researchers for their efforts in sharing raw data on OpenNeuro.org, and using open-source pipelines for analysis to maximise transparency.

There are a couple of points that made me sceptical as to whether the central claim is sufficiently robust to merit publication in its present form.

1. The main result depends upon first rejecting subjects who have a poor representation of the current state, and then only using the remaining subjects to look for representations of the future state (and interactions with repetition). This is OK in principle. However, given that one of the main results is only marginally significant ($p = 0.047$), it could leave the reader suspicious that the exclusion criteria were driven by a desire for this main result to have become significant. More details of these exclusions are needed. The most transparent way for the results to be plotted would be in figure 3B to not have error bars, but instead have individual data points for each subject. Those subjects who the authors have excluded could also be plotted but in a different colour/marker style, and the figure legend can highlight that the bar height reflects the mean of subjects who are included based upon their current state representation being above a certain threshold (the threshold and number of subjects excluded should be reported in the methods).
2. There are potential difficulties with testing decoding accuracies against chance level using T-tests (see Allefeld, Gørgen and Haynes, Neuroimage 2016). I'm not sure that this appears to be a problem for the Future*Rep interaction term (as the question here relates to whether decoding accuracy changes across the experiment), but perhaps it is a problem for the Future state decoding?

Minor comments

Results section: when the task is first introduced, it would help if we knew the core details about the number of different mazes subjects saw (8?), how far apart repetitions of mazes were on average (~4 minutes), etc. This info is already in the methods, but it would aid clarity to mention it also in the results. As part of this, I would also suggesting shifting (or copying) the sentence "This large delay strongly reduces the likelihood of between-repetition working memory maintenance as an explanation of learning performance, a common problem in reward learning paradigms where learning repetitions are separated only by several seconds on average" from the methods to the results.

p.10 For the visual ROIs, there is a typo for the AUC (it should lie between 47.61 and 48.85)

Reviewer #1

In this paper Wimmer and Buchel investigate the coding of current and predictive state information in a temporally extended Y maze using fMRI. In their task participants move through a Y maze that a number of different states. Crucially, these states include an initial choice state (State 1) and two pre-reward states (State 3) that are dependent on the choice. Each of these three states is represented by a different category of object (faces, scenes or objects) allowing past, present, future (and even counterfactual) states to potentially be decoded using MVPA techniques. Using this approach they find representations of both present and future state in OFC, and representations of present state in hippocampus. Moreover, the representation of future state in OFC gets stronger with learning.

These findings add to mounting evidence that OFC represents state information during learning and decision making tasks. The novelty in this paper is twofold: first they show how *predictive* state information is encoded in OFC over relatively long times (40s); and second they show that these predictive representations develop quickly (after 1 trial, but increasing up to 3 trials). These are important results and, I think, would be appropriate for Nature Comms. Apart from the first point below, my comments are mostly minor.

We thank the reviewer for their insightful reading of our manuscript and greatly appreciate that she or he finds our results to be both novel and important. We aim to fully address the concerns below in our response.

Perhaps my biggest reservation about this paper is the relatively small sample size (32 after exclusions I believe) that are used to detect relatively small effects. Such small effects are to be expected with a classification paper – especially in OFC – where Schuck et al. also have pretty small effects. This concern is somewhat mitigated by the strong prior hypotheses which, for example, allowed them to define an OFC ROI based on earlier work, but it still puts a question mark on the results for me. If this were a behavioral study I would ask for a replication, for an fMRI study I realize such a request may be too much. However, I would like to see the small effects acknowledged a bit more in the Discussion – perhaps even report a power analysis showing the number of subjects that would be required for a replication of the main findings. Also I think it would be good to compare the classification accuracies with Schuck et al. – are you in the same ballpark for classification accuracy of the current state?

We appreciate the reviewer's concern about the strength of the primary OFC-VMPFC future by correct repetition interaction (shown in Figure 3B) and we have

focused our work in revision on additional analyses that we believe provide significant support for this result.

We examined the strength of the OFC-VMPFC future state by correct repetition interaction and compared this representation of hidden information, as well as on-screen stimulus decoding, to a recent paper in this area (Schuck et al. 2016, Neuron). These results are described in the Results section (p. 10, p. 12) and the Discussion section (p. 21-22).

Our total scanned sample size was 35; after exclusions based on poor behavior, movement, and fMRI localizer quality, 31 participants remained. Of these, two exhibited poor generalization of classification from the localizer to the learning phase and were excluded (note this this exclusion did not affect our results; see below), leaving 29 participants in the multivariate analysis. Importantly, this number of participants for classification analyses of 'hidden' state representations in the OFC is in the same range as the number of participants in Schuck et al. 2016, which included 27 participants.

The critical OFC-VMPFC future state by correct repetition effect size is $d = 0.47$. With this medium effect size, a minimum of 38 participants would be required in a replication to achieve 80 % power to detect an effect. This sample size is within the range of common fMRI sample sizes, and indeed is only a few participants more than our initial sample size before exclusions. We now include this information in the Results (p. 12).

Regarding collecting additional fMRI data, while it would normally be difficult to collect additional participants to contribute to the sample months or years after the original acquisition, in our case, this is not a possibility, as the Siemens scanner at the UKE underwent a significant upgrade including a new gradient and RF system since the MRI data was acquired.

For the comparison of sample size, in the paper by Schuck et al. (2016), 27 participants are included in their OFC hidden state analysis. Their results for decoding current state (which was composed of both stimulus category and stimulus age) versus chance (50 %) were mixed, although exact statistics are not reported. Current state category was decoded at approximately 52 % (noted as $p < 0.05$), while current state age was below-chance, at approximately 49.5 %. Our ability to decode current state (faces, scenes, and objects) in the OFC-VMPFC in our paradigm appears to be higher, with an AUC of 9.88 (versus zero; CI [6.45 13.31]; $t_{(30)} = 5.88$, $p < 0.001$). This comparison is noted in the Results (p. 10).

The OFC-VMPFC hidden state classification results of Schuck et al. 2016 are stronger than our OFC-VMPFC future by repetition effect. It is important to consider, however, that because we focused specifically on learning across long delays, the rapid event-related experiment by Schuck et al. included more than 10x as many trials as in our design. This large difference in trial numbers likely contributes to the different effect sizes. We note this in the Discussion (p. 21-22).

We believe the additions to the manuscript provide support for our result, as well as providing a foundation for future replications.

Minor points

As I understand the design of this experiment, because the classifier is trained on a localizer task with no value, there's no confound here between expected value and future state. Is this correct? If so, I think this point should be highlighted, because the first question that comes to mind when reading this is that you are simply decoding value not state.

This is a critical point to clarify and we appreciate the reviewer bringing it up. The reviewer is correct in stating that there is no confound between value and future

state. We have added a discussion of this point to the Methods (p. 38) as well as the Results (p. 11).

Because the classifier is trained after learning and all 3 categories ended with approximately the same average value, discrimination of localizer patterns is very unlikely to be affected by value. Moreover, it is not possible for any changes in classification over learning – which underlie our OFC-VMPFC effect – to be affected by reward associations present during the localizer because the trained classifier itself does not change.

Would be nice to have a searchlight analysis for classifier. I realize this will be terribly underpowered, but (as with the exploratory PPI analysis) would give a hint of whether there are other regions involved and how specific results are to OFC.

Thank you for this suggestion. To explore the selectivity of the OFC-VMPFC effect, we have conducted an equivalent ROI analysis tiling the PFC and found only 1 of 25 regions exhibited an effect at the $p < 0.05$ level. These results are included in the Results section (p. 15; see also the Methods section p. 41-42) and in Figure S10 and Table S1.

A common approach to whole-region analyses is to use a “searchlight” analysis, based on classification accuracy in a spherical region around each voxel. Our primary multi-category and multiple regression analysis of state information representation change, however, makes a common searchlight approach quite complex. As an alternative, we examined the information content of regions in the PFC in an approximately tiled manner. This approach has a benefit over searchlight analyses in that it pools voxels together that have a common functional architecture, thus respecting the boundaries of different functional regions.

Our PFC regions of interest were selected from a 50-region whole-brain parcellation map derived from coactivation patterns across more than 10,000 published studies in the Neurosynth database (Chang et al., 2018).

From this parcellation mask, we extracted 14 ROIs (5 medial and 9 with separate left and right ROIs) in the prefrontal cortex with clustered spatial distributions and sizes approximately matching our OFC-VMPFC ROI, plus one additional self-created bilateral OFC ROI in a region lacking parcellation coverage (Figure S10); see the full ROI map here: <https://neurovault.org/images/122507/>. In addition to the PFC ROIs, we also separately re-analyzed hippocampal responses by subdividing the ROI into 4 subregions: right anterior hippocampus, left anterior hippocampus (both defined by $y > -20$), right posterior hippocampus and left posterior hippocampus (both defined by $y < -20$).

In these 29 additional analyses we find only one region in the lateralized DLPFC exhibiting a future by repetition effect ($p < 0.01$; Results p. 15). The results for state 1 are described in Table S2, which includes the strength of current state decoding across all 31 participants and the strength of the future state by correct repetition along with the number of included participants.

The lack of general effects supports our a priori focus on the OFC-VMPFC, indicating relative selectivity. Second, the selective OFC-VMPFC effect indicates that this result is not likely to be due to a general confound or weakness in the analysis and design that would produce this result frequently irrespective of an underlying true effect.

New PPI analysis:

The reviewer also mentions the exploratory psychophysiological interaction (PPI) in the original submission. To provide further support for the OFC-VMPFC effect, we conducted a new PPI analysis to examine whether changes in OFC-VMPFC

connectivity related to the increase in the coding of future state representations in the OFC-VMPFC across the task. We find an increase in OFC-VMPFC to hippocampal activity across learning that correlates with the OFC-VMPFC future by repetition effect. These results are described in the Methods section (p. 35-36) and the Results section (p. 16-17).

Our initial submission included a basic connectivity analysis examining general connectivity between the OFC-VMPFC and other brain regions across different maze states. This analysis identified strong connectivity between the OFC-VMPFC and hippocampus, but these effects were not linked to task behavior or neural information coding. We conducted a new PPI model to specifically test for learning effects. We thus examined changes in OFC-VMPFC connectivity with the rest of the brain across repetitions of mazes that were successfully learned.

The psychological term in the PPI was a contrast between repetitions 2 and 3 (late learning) and repetitions 0 and 1 (early learning). Mazes where above-chance performance was not achieved were omitted from this contrast. Our goal in this analysis was to examine changes in connectivity that may relate to learning, so we additionally entered the OFC-VMPFC future by correct repetition effect as a covariate. This analysis tests for increases in OFC-VMPFC connectivity across learning that positively relate to increases in OFC-VMPFC future state representation across learning. Additional PPI models were estimated for state 2, state 3, and the feedback state as controls.

We found no significant positive or negative main effects of learning repetition on OFC-VMPFC connectivity with other regions. Importantly, however, we found a significant positive relationship between the strength of OFC-VMPFC connectivity and individual differences in OFC-VMPFC acquisition of future state representation in a cluster including the right hippocampus, midbrain, and thalamus (Results, p. 16-17; Figure S12). This indicates that increases in OFC-

VMPFC representation of future states across learning were related to concurrent increases in functional connectivity between the OFC-VMPFC and hippocampus. Moreover, the relationship with hippocampal activity showed a clear effect not driven by outliers (Figure S12C). Additional clusters included the putamen as well as the dorsal medial and lateral PFC (Figure S12; Table S3). As a control, we conducted the same PPI analysis on the different time periods in a maze trial (state 2, state 3, and the feedback state) and found no significant correlations with the OFC-VMPFC future by repetition effect (Table S3).

Legends on the supplementary Figures would make for easier reading! Figure S1 especially, but also S3, S4, S5.

We apologize for the hard to follow figures and have now added legends.

I see that the data will be shared (thank you!), but will the analysis code also be shared? Also, will the shared data include the excl.

Thank you for this suggestion to share the multivariate data and code. We have added these to a public repository (<https://github.com/gewimmer-neuro/ofc-prediction>).

Reviewer #2

In this paper, Wimmer & Buchel examine the role of the hippocampus and OFC in acquiring predictive information during goal directed learning. Unlike many previous reports, they focus on rewards/feedback that is extended well beyond just a few seconds. They find that the hippocampus exhibited the highest activity during initial exposure; this hippocampal activation decayed over additional repetitions of the maze. In addition, they also report that patterns within the OFC/VMPFC that represent predictive information about temporally distant states around 30 seconds in the future.

Overall, I think this is a clever study and a well-written manuscript with rigorous analyses. The authors should be commended for the open sharing of data and statistical maps (and presumably analysis code after publication). Nevertheless, I have a few (relatively minor) concerns that should be addressed in a revision.

We thank the reviewer for their constructive feedback on our manuscript and greatly appreciate that she or he finds our design to be clever and our analyses to be rigorous. We aim to fully address the concerns below in our response.

Primary Issues

1) Perhaps I missed something, but the relationship between states also isn't clear. Understanding how the states (and decisions) are related could put this work into the context of decision trees (e.g., Huys et al., 2012, PLoS Computational Biology; and other work from Peter Dayan). Although these paradigms generally involve feedback at each stage, the overall final outcome is temporally extended, similar to the present work. Thus, I think it would be beneficial to discuss this work and related papers, particularly ones that omit intervening feedback.

We greatly appreciate the suggestion to broaden the discussion of our results and relate it to other work on multi-stage decision making. (In part, we limited the length of the Discussion in our first submission to fit within a brief report format.) We have added a discussion of these connections to the Introduction section (p. 5) and Discussion section (p. 18-19).

As described in response to the next comment, we have also endeavored to greatly improve our description of the task in the Introduction (p. 5) and Figure 1.

Our experimental paradigm was designed to be relatively simple in order to isolate effects of interest and enable a reasonable scan duration for the participants. This led to only including a single choice at the top of the Y-maze, as including choices at later states or more than 2 branches would exponentially increase the time that participants needed to explore each maze. (Which would, in turn, reduce the number of possible mazes and reduce our power to detect effects.)

We strongly agree that our OFC-VMPFC findings may inform studies of more complex multi-state problems or decision trees such as that described by Huys et al. 2012 and Lee et al. 2014 (Intro, p. 5; Discussion p. 18-19). Our results, in combination with studies in rodents, indicate that the OFC-VMPFC may be critical for representing goal states of potential actions as learning develops.

2) The description of the task is not completely clear to me. Complicating this matter is the fact that the journal guidelines place the Methods at the end of the paper, making it more important to have a clear task description before presenting the primary results. Thus, I think expanding the figure caption and Introduction to clarify the goals of the maze task at each stage (e.g., what is the subject doing and why) could help.

We apologize for the difficulty in understanding the task (our original submission text was aiming for the word limits of a brief report) and we greatly appreciate the suggested clarifications.

We have added a further description of the task to the Introduction (p. 5), expanded the description of the task in the caption to Figure 1, and highlighted the critical period for multivariate decoding analyses in Figure 1 with a shaded and labeled blue box above state 1.

3) Regarding the predictive OFC responses in Figure 3, are these effects time locked to the initial state? That seems to be the case, but I think it would be important to show that the results are specific to that state and not seen at other time periods and state transitions.

It is a critical point that our results are time-locked to the onset of the maze state 1, as the reviewer correctly assumes, and we appreciate the reviewer suggesting improvements to the description of this analysis. We have added this point to the Results section (p. 10-11), Methods section (p. 37), Figure 1, and the caption for Figure 3.

We indeed find that the OFC-VMPFC future (state 3) by correct repetition effect is selective to the onset of the maze trials. The effects in state 1 are the only significant positive effects of future by repetition (and this results is not affected by outlier exclusion, see below response to Reviewer 3).

Additionally, we now highlight the results at other states that are included as supplemental Figures S7-S9. Of interest, OFC-VMPFC activity at state 2 onset shows a trend for a positive future by correct repetition effect ($p = 0.062$), indicating that OFC-VMPFC representation of future states may continue for multiple seconds after the choice.

At state 3, the same analysis finds a *decrease* in representation of the state 3 stimulus across learning. This effect may indicate that as the OFC-VMPFC circuit increases expectations for state 3 across learning, actual responses to state 3 decrease (as would be expected in a predictive coding framework). Finally, at the feedback state, all responses in all ROIs are largely flat.

The selective finding of a positive OFC-VMPFC future by repetition effect time-locked to state 1 (with a weaker trend effect at state 2) supports the interpretation of this effect as a meaningful representation of future information.

Category generality:

To provide further support for the VMPFC-OFC effect at state 1, in response to the reviewer's concerns, we also examined whether OFC-VMPFC information coding was similar across the three categories. We indeed find the same pattern of results across the three categories. These results are described in the Methods section (p. 41), the Results section (p. 15), and in a new supplemental figure, Figure S5.

The regressions used in our analysis of the multivariate data collapse across the three categories in the mazes (face, scene, and object) that collectively allow for us to examine information coding during state 1 choices. As shown in Figure S5, the OFC-VMPFC shows a positive effect in the future state by correct repetition interaction, although the interaction effect for objects is weaker than the effect for faces or scenes. (The weaker effect for objects is not surprising given that decoding of on-screen object stimuli was weaker than faces or scenes in the full $n = 31$ group of participants.)

Minor Comments

1) I'm not sure I fully understand the advantage of MVPA in this paper. Could the authors arrive at the exact same conclusions with standard univariate analyses? Is there a specific pattern in the OFC that is predictive? Would that pattern be visible in standard univariate analyses? Please clarify.

We appreciate the opportunity to clarify the importance and features of the multivariate analysis, and have added a discussion of this to the Methods (p. 36). The multivariate analysis is critically important to examine information content in our regions of interest, and especially for detecting changes in representations over time.

For the internal representation of a single stimulus category, a univariate response is sometimes sufficient if categories evoke responses in different regions (e.g. Wimmer & Shohamy 2012, *Science*). However, if the goal is to understand internal or predictive representations in the same region for multiple categories, univariate responses are not sufficient.

If region A shows no mean univariate response differential between categories, then region A cannot be used to understand information representation. For example, the hippocampus shows no strong differences in univariate responses to faces, scenes, or objects. Nevertheless, from multivariate distributed patterns of voxel activity, we can accurately decode at a reasonably high level whether a given stimulus on the screen is a face, scene, or object.

However, even if a region shows a univariate response to different categories, it does not help disambiguate representational content. For example, say that region A shows a univariate increase in responses to faces, a decrease in response to scenes, and a medium positive response to objects. Over the course of learning, the participant learns to avoid the initially-selected arm that ends in a face and select the arm that ends in a scene. And in region A, we see a decrease in univariate activity at state 1. It is not possible to disambiguate whether this means that the distal scene arm is represented more across learning, that the response to the on-screen object decreases over time, or that the response to the avoided face arm decreases over time.

A univariate analysis also cannot detect the increase in future state representation over repetitions for an additional more practical reason because the classifiers are trained to discriminate categories versus other categories.

For these reasons, we did not look for (and did not find) a specific univariate response in the OFC-VMPFC that was related to our multivariate decoding results (the future by correct repetition interaction).

(It is important to note that this category-versus-category classifier training does not affect the regression results; as shown in Figure S4, the effects do not change with the inclusion of a regressor for a different category, e.g. the state 1 on-screen stimulus.)

2) The neuroimaging results are particularly noisy. Although the authors implement standard corrections for head motion, there's still activation outside the brain and in the white matter (e.g., Figure 2). Is this due to how the task is being modeled or are there potentially sources of noises that are unaccounted for (or both)? I realize there is always unaccounted for variance, but I am concerned that some of the presented results are not within grey matter and could be due to head motion (or other issues).

We appreciate the reviewer's attention to the fMRI univariate results. The reviewer is correct that several clusters extend beyond the mean anatomical image underlay. The full SPM results images can be viewed – with variable thresholding or no threshold – at <https://neurovault.org/collections/4420/>

The greater extent of some activation clusters is due to the lower mean signal intensity threshold we applied in whole-brain masking during first-level analysis. We deliberately did not employ any final stage “cosmetic” masking for appearance, as is often employed in fMRI data analyses. Our data thus might appear to be hampered by artefacts, whereas in fact we simply show unmasked data.

SPM by default excludes regions below a threshold of 80 % of mean across-volume signal, which usually leads to results restricted only to the brain. Critically,

depending on the ratio between brain and background voxels this can potentially remove signal in brain regions with low signal intensity such as the ventral striatum, medial temporal lobe, and orbitofrontal cortex from the results. (E.g. early studies in the 2000s of reward prediction error using SPM did not find results in the ventral striatum potentially because of this setting (O’Doherty et al. 2003, *Neuron*).)

In order to examine results in these regions, and as a consequence of not using this high cut-off, our results maps also include lower mean signal voxels outside of cortical grey matter, which, due to normal smoothing (6mm FWHM) leads to some effects “bleeding over” into the space outside the average anatomical, as the reviewer points out. However, after preprocessing, we did apply a dilated mean anatomical mask to the functional data in order to limit multiple comparisons correction at the second level.

Effects extending beyond the edge of the brain would be seen in any unmasked fMRI study with smoothing and across-participant anatomical variability. For many studies not focused on low-SNR regions, the high 80 % signal cutoff employed by SPM indeed leads to a clearer exclusion of voxels outside the cortex and the appearance of cleaner results. This would give the impression that our results are outliers, but this impression is due to the particular features of the common analysis pathway described above.

The distribution of our fMRI results in the current study are similar to other studies where we have used similar methods to examine low-signal regions, including the results of Wimmer & Shohamy, 2012 *Science* (<https://neurovault.org/collections/2389/>), and this spread can be found across other studies where full maps are uploaded, such as Chang et al. 2016 *PLoS Biology* (<https://neurovault.org/images/1696/>).

We are happy to present our results in full and without additional masking (<https://neurovault.org/collections/4420/>), and additionally provide the full fMRI dataset at openneuro.org for those who wish to replicate our analyses. If it is helpful, we would be happy to add a statement about the above to the current manuscript.

3) I think it would be helpful for the authors to further clarify the relationship between the motivation for this paper and their recent Journal of Neuroscience paper (“Reward learning over weeks versus minutes increases the neural representation of value in the human brain”). Although the papers seem to be focused on distinct questions, both seem to address the temporal nature and maintenance of learning signals.

Thank you for pointing out the relationship between our current paper and our recent publication (*J Neurosci*, 2018). We have added a discussion of this point to the Discussion section (p. 20).

Our recent paper focused on learning from relatively immediate feedback but where learning sessions were spaced over weeks versus condensed into one session.

In that paper, spacing between trials within a session was relatively condensed and similar to other rapid event-related designs. Our current paper is focused instead on examining predictive representations during learning where feedback is significantly delayed, by more than 40 seconds.

However, there is some broad conceptual overlap regarding between-learning-event spacing in that repetitions of mazes are spread across more than 4 min, while sessions in our previous study are spread across days in the recent study.

Additionally, as in the current study, we found a role for working memory in the maintenance of recently learned value associations.

4) In Figure 3, what is chance in the predictions? The classification accuracy seems quite low, but the authors don't provide a clear intuition about the nature of chance in these predictions.

We apologize for the lack of a specific definition of chance for the critical regression analysis in our manuscript and have added a discussion of this to the Methods (p. 39) and Results (p. 11).

Our regression analysis on decision values on the single participant level provides regression coefficients for the main effects and interactions (with individual participant effects now shown as black points in Figure 3B and Figure S3). At the second level, we conduct one-sample t-tests (two-tailed) across these coefficients to determine whether they are significantly different from zero. Thus, while our primary analysis does not rely on typical multivariate measures of chance (e.g. 50 %) and should not be interpreted as a percentage measure, the effect size of $d = 0.47$ that we now include in the revised manuscript can be compared across studies.

Further, the critical effect in the OFC-VMPFC is related to a change in representations across learning; thus, the comparison is not to chance or zero but to the change in this effect across four repetitions (as shown in Figure 3C).

For the basic initial sanity check of across-phase classification of face, scenes, and objects, we use AUC, where the chance level is also zero (similar to a 50 % chance classification baseline). We have added a note (Results, p. 10) that our results in the OFC-VMPFC are comparable or higher than other studies (e.g. Schuck et al. 2016).

Reviewer #3

This paper examines whether a representation of a forthcoming state can be decoded from human orbitofrontal cortex when deciding between two alternative courses of action. This question is of interest because of the recent literature suggesting that the hippocampus and orbitofrontal cortex may together represent the structure of the environment, in the form of a cognitive map.

The central result of the paper (figure 3) is that when choosing between two actions at an initial starting state ('state 1'), it is possible to use multivariate pattern analysis on OFC/VMPFC data to: (a) not only decode whether subjects are currently examining a face, house or scene ('current state'); (b) but also whether they will examine a face/house/scene in an intermediate state ('state 3') visited en route to reward, which is modulated by how often that state has been viewed in the past ('future * repetition' interaction).

A strength of the study is that it cleverly examines how a map may be learnt over periods of time that are less susceptible to being solved using working memory (maze repetitions are several minutes apart). It also looks at predictive information over a longer timescale typically than that studied in reinforcement learning tasks. If the claim is robust, it will be of interest to researchers interested in memory, learning and decision making. I also applaud the researchers for their efforts in sharing raw data on OpenNeuro.org, and using open-source pipelines for analysis to maximise transparency.

We thank the reviewer for their insightful reading of our manuscript and greatly appreciate that she or he finds our question of interest and our results to be of broad interest to the neuroscience community. We aim to fully address the concerns below in our response.

There are a couple of points that made me sceptical as to whether the central claim is sufficiently robust to merit publication in its present form.

1. The main result depends upon first rejecting subjects who have a poor representation of the current state, and then only using the remaining subjects to look for representations of the future state (and interactions with repetition). This is OK in principle. However, given that one of the main results is only marginally significant ($p = 0.047$), it could leave the reader suspicious that the exclusion criteria were driven by a desire for this main result to have become significant. More details of these exclusions are needed. The most transparent way for the results to be plotted would be in figure 3B

to not have error bars, but instead have individual data points for each subject. Those subjects who the authors have excluded could also be plotted but in a different colour/marker style, and the figure legend can highlight that the bar height reflects the mean of subjects who are included based upon their current state representation being above a certain threshold (the threshold and number of subjects excluded should be reported in the methods).

The criteria for inclusion and the robustness of our results is a critical point and we welcome the reviewer's questions in this area. In short, we find no influence on the OFC-VMPFC future by correct repetition effect of the inclusion of 2 excluded participants. These findings are described in the Results (p. 14) and represented in Figure S3, and Figure 3B has been updated to show individual datapoints from included participants in a violin plot.

We found no influence on the OFC-VMPFC future by correct repetition effect of the inclusion of 2 excluded participants. The 2 excluded participants did not meet the a priori requirement of above-zero classification of actual on-screen state 1 and state 3 stimuli in any of the 3 categories. When included, across In the full group of 31 participants, the OFC-VMPFC effect remains significant ($p = 0.027$; Results, p. 14). The results including all participants are shown in Figure S3, which highlights the 2 excluded participants with yellow dots (black dots represent the other participants). A separate panel in Figure S3 includes the new main Figure 3B violin plot for direct comparison. We believe these results further support the robustness of the OFC-VMPFC effect.

Regarding the main effect of future state in the OFC-VMPFC, which the reviewer correctly notes is weaker effect than the critical future state by correct repetition interaction, we did not have specific a priori predictions. It is not clear what an effect of future state alone means, given that this analysis includes the initial repetition where the future category has not yet been experienced. (Indeed, this effect is not significant when including all participants as shown in Figure S3.) We

did not intend to highlight this result and have adjusted the text accordingly (Results, p. 12).

2. There are potential difficulties with testing decoding accuracies against chance level using T-tests (see Allefeld, Gørgen and Haynes, Neuroimage 2016). I'm not sure that this appears to be a problem for the Future*Rep interaction term (as the question here relates to whether decoding accuracy changes across the experiment), but perhaps it is a problem for the Future state decoding?

We appreciate the point about significance in classification as highlighted by Allefeld et al. 2016, and while we did indeed consider it when we were examining our results, we apologize for omitting a discussion of how this concern relates to our data. The reviewer is correct that the primary finding, the future state by correct repetition interaction in the OFC-VMPFC, is not a test against any chance or null-information value and thus does not fall under the area of concern highlighted by Allefeld et al. This is addressed in the revision (Methods p. 40; Results p. 11-12).

To explicitly address the concerns about classification versus chance, we have added a discussion to the Methods section (p. 40) where we specify that our primary predictions are not affected by this issue. Further, specifically for the understanding of current state classification in the Results (p. 11-12), we now add a caveat about interpreting the strength of these effects, noting, however, that we are testing cross-classification and not relying on the cross-validation case which presents the specific problem (as explicitly stated in Allefeld et al. 2016).

The basic concern raised by Allefeld et al. is that second-level t-tests on cross-validated information measures do not provide for population inference. The consequence is that t-tests on such results support fixed-rather than random-effects inference.

We agree with the reviewer that the primary finding, the future by repetition interaction, is not a test against any chance or null-information value and thus does not fall under the area of concern highlighted by Allefeld et al. Specifically, we do not rely on classification evidence being judged versus chance or any other value but instead of whether this evidence changes over time – a measure derived from classification values being compared to other classification values over repetitions). Thus the main claims of our paper are not affected.

Inferences about current state (and future state) classification alone are tested versus chance, and while Allefeld et al. state specifically that their argument does not apply to cross-classification cases such as ours, it is possible that these analyses are also susceptible to the concerns raised in Allefeld et al.

It is of some interest that our regions of interest exhibited above-chance discrimination of the current state, but for the future state we had no prediction about and draw no conclusions from these results. However, at the group level for the current state and future state, we do report the strength of classification using a t-test we now add a caveat to the Results (p. 11-12) about interpreting the strength of these effects alone.

Finally, for current state classification, our per-participant per-category exclusion criteria do rely on using the strength of current state classification, but this operation is reliant only on whether classification is above zero (specifically, whether the regression coefficient of trial-by-trial decision values versus actual category is greater than zero) and not on any kind significance threshold.

Minor comments

Results section: when the task is first introduced, it would help if we knew the core details about the number of different mazes subjects saw (8?), how far apart repetitions of mazes were on average (~4 minutes), etc. This info is already in the methods, but it would aid clarity to mention it also in the results. As part of this, I would also suggesting shifting (or copying) the sentence “This large delay strongly reduces the likelihood of between-repetition working memory maintenance as an explanation of learning performance, a common problem in reward learning paradigms where learning repetitions are separated only by several seconds on average” from the methods to the results.

We apologize for the difficulty in understanding the task (our original submission text was aiming for the word limits of a brief report) and we greatly appreciate the suggested clarifications.

We have added a further description of the task to the Introduction (p. 5), including the important point that repetitions were separated by several minutes which we also note in the Results (p. 7). We have also expanded the description of the task in the caption to Figure 1.

p.10 For the visual ROIs, there is a typo for the AUC (it should lie between 47.61 and 48.85)

We appreciate the detection of this typo and have added the correct AUC value (48.23).

REVIEWERS' COMMENTS:

Reviewer #1 (Remarks to the Author):

You have answered all of my concerns. Congratulations on a great paper!

Reviewer #2 (Remarks to the Author):

The authors have addressed my original concerns. The task description is much clearer and the paper is easier to follow. I only have a couple of minor follow up questions/comments that do not require re-review:

1) In reply to Reviewer 1, the authors quote an effect size of $d = 0.47$ and state that 38 participants would be needed to detect that effect with 80% power. Could the authors clarify two things: 1) What is the variability associated with this Cohen's d estimate? 2) Is this power analysis set for an alpha of $p = 0.05$ (uncorrected)?

2) I appreciate the authors' efforts to be more transparent with respect to their results by not applying additional "cosmetic" masking. And I generally follow the authors' reasoning behind the non-gray matter activation. It's true that smoothing could cause some minor bleeding over to non-gray matter voxels (e.g., air, CSF, white matter), but there must be some signal in these non-gray matter regions to start with, which is why I raised this (minor) issue originally. I think it merits a brief mention (if there's space) in the Discussion since some of these patterns of activation would be excluded with some denoising approaches (e.g., Pruim et al., 2015, NeuroImage) and newer surface- and gray matter-centric data formats (e.g., Glasser et al., 2016, Nature Neuroscience).

I thank the authors for their careful revisions and thoughtful paper. Although the current paper could potentially benefit from additional data (like nearly all neuroscience studies), the fact that the authors have openly shared their data and code raises the possibility that other research groups could easily extend this study.

Reviewer #3 (Remarks to the Author):

I thank the authors for their responses to my previous reviews. I think they have been satisfactorily addressed, and the paper now merits publication.

Reviewer #2

The authors have addressed my original concerns. The task description is much clearer and the paper is easier to follow. I only have a couple of minor follow up questions/comments that do not require re-review:

1) In reply to Reviewer 1, the authors quote an effect size of $d = 0.47$ and state that 38 participants would be needed to detect that effect with 80% power. Could the authors clarify two things: 1) What is the variability associated with this Cohen's d estimate? 2) Is this power analysis set for an alpha of $p = 0.05$ (uncorrected)?

We appreciate the opportunity to add detail to the power analysis results. The effect size of $d = 0.47$ has a confidence interval (derived from the confidence interval of the underlying effect) ranging from 0.01 to 0.85. Second, the power analysis was for a two-tailed (uncorrected) test. A one-tailed test indicates a lower sample of $n = 30$. This information has been added to the Results, p. 12-13:

“With a medium effect size ($d = 0.47$ CI [0.01 0.85]), a replication of the OFC-VMPFC result aiming for 80 % power to detect an effect would require a minimum of 38 participants (two-tailed, uncorrected; $n = 30$, one-tailed).”

2) I appreciate the authors' efforts to be more transparent with respect to their results by not applying additional "cosmetic" masking. And I generally follow the authors' reasoning behind the non-gray matter activation. It's true that smoothing could cause some minor bleeding over to non-gray matter voxels (e.g., air, CSF, white matter), but there must be some signal in these non-gray matter regions to start with, which is why I raised this (minor) issue originally. I think it merits a brief mention (if there's space) in the Discussion since some of these patterns of activation would be excluded with some denoising approaches (e.g., Pruim et al., 2015, NeuroImage) and newer surface- and gray matter-centric data formats (e.g., Glasser et al., 2016, Nature Neuroscience).

We agree that it is possible that the effects in non-gray matter regions could be due to effects of non-interest such as motion or other noise sources. For the benefit of future research projects, we have added a point on this and the relevant references provided by the reviewer.

We believe it was best to address this concern directly in the Results section (p. 8-9):

“Note that these results revealed some clusters of apparent activation on the edge of the grey matter which could be due to residual noise or motion; it is possible that such effects could be removed with more advanced motion- and noise-correction algorithms (Pruim et al., 2015; Glasser et al., 2016).”

I thank the authors for their careful revisions and thoughtful paper. Although the current paper could potentially benefit from additional data (like nearly all neuroscience studies), the fact that the authors have openly shared their data and code raises the possibility that other research groups could easily extend this study.

We thank the reviewers for their focus on improving the paper by supporting the main results, and by the suggestion to post the multivariate data and code online. We too hope that this will support future replications and explorations of these questions.